# CRITICAL SAMPLING FOR ROBUST EVOLUTION BEHAVIOR LEARNING OF UNKNOWN DYNAMICAL SYSTEMS

## ABSTRACT

We study the following new and important problem: *given an unknown dynamical system, what is the minimum number of samples needed for effective learning of its governing laws and accurate prediction of its future evolution behavior, and how to select these critical samples*? In this work, we propose to explore this problem based on a design approach. Specifically, starting from a small initial set of samples, we adaptively discover and collect critical samples to achieve increasingly accurate learning of the system evolution. One central challenge here is that we do not know the network modeling error of the ground-truth system state, which is however needed for critical sampling. To address this challenge, we introduce a multi-step reciprocal prediction network where a forward evolution network and a backward evolution network are designed to learn and predict the temporal evolution behavior in the forward and backward time directions, respectively. Very interestingly, we find that the desired network modeling error is highly correlated with the multi-step reciprocal prediction error. More importantly, this multi-step reciprocal prediction error can be directly computed from the current system state without knowing the ground-truth or data statistics. This allows us to perform a dynamic selection of critical samples from regions with high network modeling errors and develop an adaptive sampling-learning method for dynamical systems. To achieve accurate and robust learning from this small set of critical samples, we introduce a joint spatial-temporal evolution network which incorporates spatial dynamics modeling into the temporal evolution prediction for robust learning of the system evolution operator with few samples. Our extensive experimental results demonstrate that our proposed method is able to dramatically reduce the number of samples needed for effective learning and accurate prediction of evolution behaviors of unknown dynamical systems by up to hundreds of times, especially for high-dimensional dynamical systems.

## 1 INTRODUCTION

Recently, learning-based methods for complex and dynamic system modeling have become an important area of research in machine learning. The behaviors of dynamical systems in the physical world are governed by their underlying physical laws (Bongard & Lipson, 2007; Schmidt & Lipson, 2009). In many areas of science and engineering, ordinary differential equations (ODEs) and partial differential equations (PDEs) play important roles in describing and modeling these physical laws (Brunton et al., 2016; Raissi, 2018; Long et al., 2018; Chen et al., 2018; Raissi et al., 2019; Qin et al., 2019). In recent years, data-driven modeling of unknown physical systems from measurement data has emerged as an important area of research. There are two major approaches that have been explored. The first approach typically tries to identify all the potential terms in the unknown governing equations from a priori dictionary, which includes all possible terms that may appear in the equations (Brunton et al., 2016; Schaeffer & McCalla, 2017; Rudy et al., 2017; Raissi, 2018; Long et al., 2018; Wu & Xiu, 2019; Wu et al., 2020; Xu & Zhang, 2021). The second approach for data-driven learning of unknown dynamical systems is to approximate the evolution operator of the underlying equations, instead of identifying the terms in the equations (Qin et al., 2019; Wu & Xiu, 2020; Qin et al., 2021a; Li et al., 2021b).

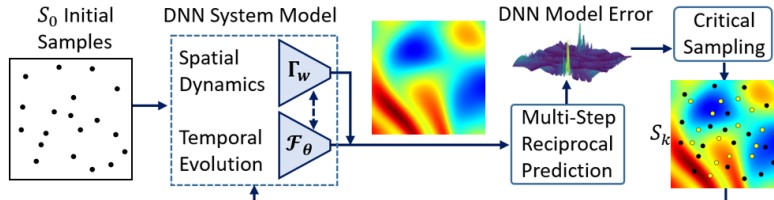

Figure 1: Illustration of the proposed method of critical sampling for accurately learning the evolution behaviors of unknown dynamical systems.

Many existing data-driven approaches for learning the evolution operator typically assume the availability of sufficient data, and often require a large set of measurement samples to train the neural network, especially for high-dimensional systems. For example, to effectively learn a neural network model for the 2D Damped Pendulum ODE system, existing methods typically need more than 10000 samples to achieve sufficient accuracy (Qin et al., 2019; Wu & Xiu, 2020). This number increases dramatically with the dimensions of the system. For example, for the 3D Lorenz system, the number of needed samples used in the literature is often increased to one million. We recognize that, in practical dynamical systems, such as ocean, cardiovascular and climate systems, it is very costly to collect observation samples. This leads to a new and important research question: *what is the minimum number of samples needed for robust learning of the governing laws of an unknown system and accurate prediction of its future evolution behavior?*

Adaptive sample selection for network learning, system modeling and identification has been studied in the areas of active learning and optimal experimental design. Methods have been developed for global optimization of experimental sequences (Llamosi et al., 2014), active data sample generation for time-series learning and modeling (Zimmer et al., 2018), Kriging-based sampling method for learning spatio-temporal dynamics of systems (Huang et al., 2022), adaptive training of physics-informed deep neural networks (Zhang & Shafieezadeh, 2022), and data-collection scheme for system identification (Mania et al., 2022). However, within the context of deep neural network modeling of unknown dynamical systems, the following key challenging issues have not been adequately addressed: (1) how to characterize and estimate the prediction error of the deep neural networks? (2) Based on this error modeling, how to adaptively select the critical samples and successfully train the deep neural networks from these few samples?

Figure 1 illustrates the proposed method of critical sampling for accurately learning the evolution behaviors of unknown dynamical systems. We start with a small set of initial samples, then iteratively discover and collect critical samples to obtain more accurate network modeling of the system. During critical sampling, the basic rule is to select the samples from regions with high network modeling errors so that these selected critical samples can maximally reduce the overall modeling error. However, the major challenge here is that we do not know network modeling error, i.e., the difference between the system state predicted by the network and the ground-truth which is not available for unknown systems. To address this challenge, we establish a multi-step reciprocal prediction framework where a forward evolution network and a backward evolution network are designed to learn and predict the temporal evolution behavior in the forward and backward time directions, respectively. Our hypothesis is that, if the forward and backward prediction models are both accurate, starting from an original state $A$, if we perform the forward prediction for $K$ times and then perform the backward prediction for another $K$ times, the final prediction result $\bar{A}$ should match the original state $A$. The error between $\bar{A}$ and $A$ is referred to as the *multi-step reciprocal prediction error*.

Very interestingly, we find that the network modeling error is highly correlated with the multi-step reciprocal prediction error. Note that multi-step reciprocal prediction error can be directly computed from the current system state, without the need to know the ground-truth system state. This allows us to perform a dynamic selection of critical samples from regions with high network modeling errors and develop an adaptive learning method for dynamical systems. To effectively learn the system evolution from this small set of critical samples, we introduce a joint spatial-temporal evolution network structure which couples spatial dynamics learning with temporal evolution learning. Our extensive experimental results demonstrate that our proposed method is able to dramatically reduce the number of samples needed for effective learning and accurate prediction of evolution behaviors of unknown dynamical systems. This paper has significant impacts in practice since collecting samples from real-world dynamical systems can be very costly or limited due to resource/labor constraints or experimental accessibility.

The **major contributions** of this work can be summarized as follows. (1) We have successfully developed a multi-step reciprocal prediction approach to characterize the prediction errors in deep neural network modeling of unknown dynamical systems. We have made an interesting finding that the network modeling error is highly correlated with the multi-step reciprocal prediction error, which enables us to develop the critical sampling method. (2) We have designed a joint spatial-temporal evolution network which is able to learn the system evolution with few critical samples. (3) Our proposed method is able to dramatically reduce the number of needed samples and related sample collection costs for learning the system evolution, which is highly desirable in practical applications.

## 2 RELATED WORK

In this section, we review existing research closely related to our work.

**(1) Data-driven modeling of unknown physical systems.** There are two major approaches that have been explored in the literature. The first approach aims to learn the mathematical formulas or expressions of the underlying governing equations. In a series developments of this direction, the seminal work was made by (Bongard & Lipson, 2007; Schmidt & Lipson, 2009), where symbolic regression was proposed to learn nonlinear dynamic systems from data. Later, more approaches have been proposed in this direction, including but not limited to sparse regression (Brunton et al., 2016; Schaeffer & McCalla, 2017; Rudy et al., 2017), neural networks (Long et al., 2018; Raissi, 2018; Long et al., 2019), Koopman theory (Brunton et al., 2017), Gaussian process regression (Raissi et al., 2017), model selection approach (Mangan et al., 2017), classical polynomial approximations (Wu & Xiu, 2019; Wu et al., 2020), genetic algorithms (Xu et al., 2020; Xu & Zhang, 2021; Xu et al., 2021), and linear multi-step methods (Keller & Du, 2021), etc.

The second approach aims to approximate the evolution operator of the underlying dynamical system typically via a deep neural network, which predicts the system state for the next time instance from the current state (Wu & Xiu, 2020; Qin et al., 2021a; Li et al., 2021a;b). In fact, the idea of such an approach is essentially equivalent to learning the integral form of the underlying unknown differential equations (Qin et al., 2019). The performance of this approach has been demonstrated for learning ODEs (Qin et al., 2019) and modeling PDEs in generalized Fourier spaces (Wu & Xiu, 2020) and physical space (Chen et al., 2022). Recently, this approach has also been extended to data-driven modeling of parametric differential equations (Qin et al., 2021b), non-autonomous systems (Qin et al., 2021a), partially observed systems (Fu et al., 2020), biological models (Su et al., 2021), and model correction (Chen & Xiu, 2021). For an autonomous dynamical system, its evolution operator completely characterizes the system evolution behavior. Researchers have demonstrated that the evolution operator, once successfully learnt, can be called repeatedly to predict the evolution behaviors of the unknown dynamical systems (Qin et al., 2019; Wu & Xiu, 2020; Chen et al., 2022).

**(2) State-space models and adjoint state methods.** State-space models have shown to be a powerful tool for modeling the behaviors of dynamical systems (McGoff et al., 2015). Methods have been developed for approximating dynamical systems with hidden Markov model (HMM) (Rabiner, 1989; Fraser, 2008), recurrent neural network (RNN) (Han et al., 2004), long short-term memory network (LSTM) (Vlachas et al., 2018), reservoir computing (RC) (Inubushi & Yoshimura, 2017), structured variational autoencoder (SVAE) (Johnson et al., 2016), linear dynamical system (LDS) and its variations (Fox et al., 2008; Linderman et al., 2017; Gao et al., 2016).

We recognize that the multi-step forward and backward processes and the usage of mismatch errors are related to those in the recent adjoint state methods for neural ODE learning (e.g. Chen et al. (2018); Zhuang et al. (2020a;b)). However, our method is uniquely different in the following aspects. (1) In the adjoint state method, the back propagation is used to compute gradients based on a Lagrangian functional. However, in our method, the backward network is used to learn the inverse of the forward evolution operator (namely, backward evolution operator, see Lemma C.1). (2) The adjoint state method aims to compute the gradients of the loss functions more efficiently and accurately. However, our method aims to discover the critical samples for network learning.

## 3 METHOD

In this section, we present our method of critical sampling for accurate learning of the evolution behaviors for unknown dynamical systems.

### 3.1 PROBLEM FORMULATION AND METHOD OVERVIEW

In this work, we focus on learning the evolution operator $\mathbf{\Phi}_\Delta : \mathbb{R}^n \to \mathbb{R}^n$ for autonomous dynamical systems, which maps the system state from time $t$ to its next state at time $t + \Delta$: $\mathbf{u}(t + \Delta) = \mathbf{\Phi}_\Delta(\mathbf{u}(t))$. It should be noted that, for autonomous systems, this evolution operator $\mathbf{\Phi}_\Delta$ remains invariant over time. It only depends on the time difference $\Delta$. For an autonomous system, its evolution operator completely characterizes the system evolution behavior (Qin et al., 2019; Wu & Xiu, 2020; Chen et al., 2022).

Our goal is to develop a deep neural network method to accurately learn the evolution operator and robustly predict the long-term evolution of the system using a minimum number of selected critical samples. Specifically, to learn the system evolution over time, the measurement samples for training the evolution network are collected in the form of pairs. Each pair represents two solution states along the evolution trajectory at time instances $t$ and $t + \Delta$. For simplicity, we assume that the start time is $t = 0$. Using a high-accuracy system solver, we generate $J$ system state vectors $\{\mathbf{u}^j(0)\}_{j=1}^J$ at time 0 and $\{\mathbf{u}^j(\Delta)\}_{j=1}^J$ at time $\Delta$ in the computational domain $D$. Thus, the training samples are given by

$$\mathcal{S}_F = \{[\mathbf{u}^j(0) \to \mathbf{u}^j(\Delta)] : \mathbf{u}^j(0), \mathbf{u}^j(\Delta) \in \mathbb{R}^n, 1 \le j \le J\}. \tag{1}$$

It is used to train the forward evolution network $\mathcal{F}_\theta$ which approximates the forward evolution operator $\mathbf{\Phi}_\Delta$. As discussed in Section 1, we introduce the idea of backward evolution operator $\mathbf{\Psi}_\Delta : \mathbb{R}^n \to \mathbb{R}^n$, $\mathbf{u}(0) = \mathbf{\Psi}_\Delta(\mathbf{u}(\Delta))$. The original training samples in $\mathcal{S}_F$ can be switched in time to create the following sample set

$$\mathcal{S}_G = \{[\mathbf{u}^j(\Delta) \to \mathbf{u}^j(0)] : 1 \le j \le J\}, \tag{2}$$

which is used to train the backward evolution network $\mathcal{G}_\vartheta$. The forward and backward evolution networks, $\mathcal{F}_\theta$ and $\mathcal{G}_\vartheta$, form a reciprocal prediction loop.

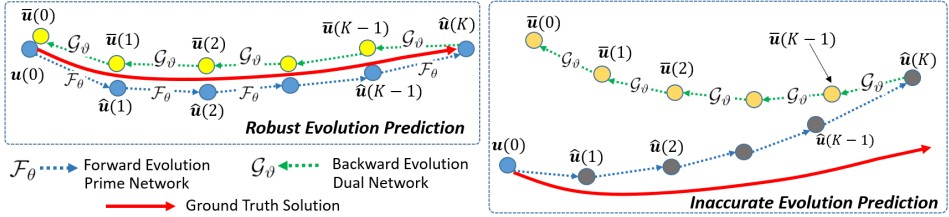

Figure 2: Illustration of the proposed idea of multi-step reciprocal prediction error.

Based on the forward and backward prediction errors, we introduce a multi-step reciprocal prediction process. As illustrated in Figure 2, starting from the initial condition $\mathbf{u}(0) \in \mathbb{R}^n$ at time $t = 0$, we perform $K$-step prediction of the system state by repeatedly calling the forward evolution network $\mathcal{F}_\theta$ with $\hat{\mathbf{u}}((k+1)\Delta) = \mathcal{F}_\theta[\hat{\mathbf{u}}(k\Delta)]$ and $\hat{\mathbf{u}}(0) = \mathbf{u}(0)$. We then apply the backward evolution network $\mathcal{G}_\vartheta$ to perform $K$-step backward prediction: $\bar{\mathbf{u}}((k-1)\Delta) = \mathcal{G}_\vartheta[\bar{\mathbf{u}}(k\Delta)]$ and get back to the initial condition $\bar{\mathbf{u}}(0)$. This process of forward and backward evolution prediction is referred to as *multi-step reciprocal prediction*. The difference between the original value $\mathbf{u}(0)$ and the final prediction $\bar{\mathbf{u}}(0)$, namely, $\mathbb{E}[\mathbf{u}(0)] = \|\mathbf{u}(0) - \bar{\mathbf{u}}(0)\|$, is referred to as the *multi-step reciprocal prediction error* in the Euclidean norm $\|\|$. In this work, we have the following interesting finding: there is a very strong correlation between the network modeling error $\mathcal{E}(\mathbf{u}(0))$ and the multi-step reciprocal prediction error $\mathbb{E}(\mathbf{u}(0))$. This allows us to use $\mathbb{E}(\mathbf{u}(0))$ to approximate the desired network modeling error $\mathcal{E}[\mathbf{u}(0)] = \|\mathcal{F}_\theta[\mathbf{u}(0)] - \mathbf{\Phi}_\Delta(\mathbf{u}(0))\|$. Note that this multi-step reciprocal prediction error can be computed directly with the current state and the forward-backward evolution networks. Its computation does not require the ground-truth system state. Therefore, we can use reciprocal prediction error to guide the selection of critical samples from regions with large modeling errors.

Let $J_m$ be the number of samples at $m$-th iteration of our critical sampling process, current sample set $\mathcal{S}_F^m = \{[\mathbf{u}^j(0) \to \mathbf{u}^j(\Delta)] : 1 \le j \le J_m\}$ is used to train the spatial-temporal evolution network. The corresponding forward-backward evolution networks are denoted by $\mathcal{F}_\theta^m$ and $\mathcal{G}_\vartheta^m$. Using the corresponding multi-step reciprocal prediction error distribution $\mathbb{E}(\mathbf{u}(0))$, we can determine regions with high error values and collect a new set of samples $\mathbf{\Omega}^m$, which are added to the existing set of samples to update the training set:

$$\mathcal{S}_F^{m+1} = \mathcal{S}_F^m \bigcup \mathbf{\Omega}^m = \{[\mathbf{u}^j(0) \to \mathbf{u}^j(\Delta)] : 1 \le j \le J_{m+1}\}. \tag{3}$$

The above sampling-learning process is repeated until the overall prediction error drops below the target threshold. According to Table 3 in Supplemental Materials, the complexity for training is increased by 4-5 times, which directly depends on the number of iterations needed to reach the threshold for the network modeling error. However, the above critical sample selection process can dramatically reduce the number of needed training samples and related sample collection cost. In the following sections, we will explain this process in more details.

### 3.2 MULTI-STEP RECIPROCAL PREDICTION ERROR AND CRITICAL SAMPLING

In this section, we show that there is a strong correlation between the multi-step reciprocal prediction error and the network modeling error of the temporal evolution network $\mathcal{F}_\theta^m$.

**(1) Multi-step reciprocal prediction.** In our multi-step reciprocal prediction scheme, we have a forward temporal evolution network $\mathcal{F}_\theta^m$ and a backward evolution network $\mathcal{G}_\vartheta^m$, which model the system evolution behaviors in the forward and backward time directions. If the forward and backward evolution networks are both well-trained, accurately approximating the forward and backward evolution operators, for an arbitrarily given system state $\mathbf{u}(0)$, the reciprocal prediction error $\mathbb{L}_S = \|\mathbf{u}(0) - \mathcal{G}_\vartheta^m[\mathcal{F}_\theta^m[\mathbf{u}(0)]]\|$ should approach 0. Now, we extend this one-step reciprocal prediction to $K$ steps. As illustrated in Figure 2, starting from the initial condition $\mathbf{u}(0)$, we repeatedly apply the forward evolution network $\mathcal{F}_\theta^m$ to perform $K$-step prediction of the system future states, $\hat{\mathbf{u}}(k\Delta) = \mathcal{F}_\theta^{m,(k)}[\mathbf{u}(0)]$, where $\mathcal{F}_\theta^{m,(k)}$ represents the $k$-fold composition of $\mathcal{F}_\theta^m$:

$$\mathcal{F}_\theta^{m,(k)} = \underbrace{\mathcal{F}_\theta^m \circ \mathcal{F}_\theta^m \circ \cdots \circ \mathcal{F}_\theta^m}_{k-\text{fold}}. \tag{4}$$

After $K$ steps of forward evolution prediction, then, starting with $\hat{\mathbf{u}}(K\Delta)$, we perform $K$ steps of backward evolution prediction using network $\mathcal{G}_\vartheta^m$: $\bar{\mathbf{u}}(k\Delta) = \mathcal{G}_\vartheta^{m,(K-k)}[\hat{\mathbf{u}}(K\Delta)], k = K - 1, \cdots, 1, 0$, where

$$\mathcal{G}_\theta^{m,(K-k)} = \underbrace{\mathcal{G}_\theta^m \circ \mathcal{G}_\theta^m \circ \cdots \circ \mathcal{G}_\theta^m}_{(K-k)-\text{fold}} \tag{5}$$

and reach back to time $t = 0$. If the forward and backward evolution networks are both accurate, the forward prediction path and the backward prediction path should match with each other. Motivated by this, we define the multi-step reciprocal prediction error for the forward evolution network $\mathcal{F}_\theta^m$ as the deviation between the forward and backward prediction paths:

$$\mathbb{E}[\mathbf{u}(0)] = \sum_{k=0}^{K} \left\| \hat{\mathbf{u}}(k\Delta) - \bar{\mathbf{u}}(k\Delta) \right\|^2. \tag{6}$$

Note that, when computing $\mathbb{E}[\mathbf{u}(0)]$, we only need the current system state $\mathbf{u}(0)$, the forward and backward evolution networks $\mathcal{F}_\theta^m$ and $\mathcal{G}_\vartheta^m$. Figure 3 shows several examples from the Damped Pendulum and 2D Nonlinear ODE systems listed in Table 2 in Supplemental Materials. The top row shows examples with accurate prediction of their system states. We can see that their forward and backward prediction paths match well and the corresponding multi-step prediction error is very small. For comparison, the bottom shows examples with large prediction errors.

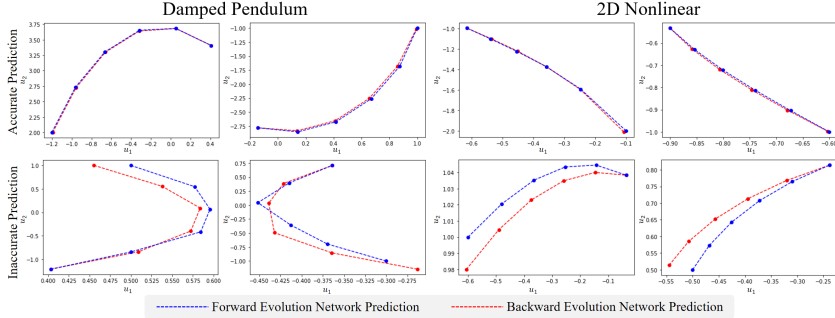

Figure 3: Examples of multi-step reciprocal prediction errors on the first two dynamical systems.

**(2) Predicting the network modeling error.** In this work, we find that there is a strong correlation between the network modeling error $\mathcal{E}[\mathbf{u}(0)]$ and the multi-step reciprocal prediction error $\mathbb{E}[\mathbf{u}(0)]$. Figure 4 shows four examples of $\mathbb{E}[\mathbf{u}(0)]$ (top row) and $\mathcal{E}[\mathbf{u}(0)]$ (middle row) for the Damped

Pendulum and 2D Nonlinear system with different sizes of training samples. The bottom row shows the values of $\mathcal{E}[\mathbf{u}(0)]$ and $\mathbb{E}[\mathbf{u}(0)]$ of locations with large errors. We can see that there is a strong correlation between the network modeling error $\mathcal{E}[\mathbf{u}(0)]$ and the multi-step reciprocal prediction error $\mathbb{E}[\mathbf{u}(0)]$. This correlation allows us to predict $\mathcal{E}[\mathbf{u}(0)]$ using $\mathbb{E}[\mathbf{u}(0)]$ which can be computed directly from the current system state without the need to know the ground-truth state.

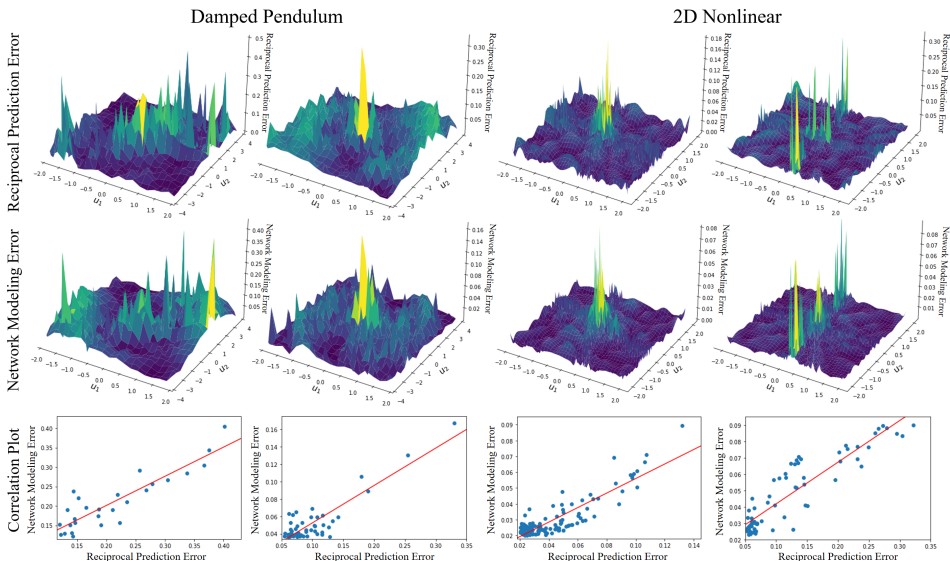

Figure 4: Correlation between network modeling error and multi-step reciprocal prediction error on Damped Pendulum and 2D Nonlinear ODE systems.

**(3) Critical sampling and adaptive evolution operator learning.** Once we are able to predict the network modeling error $\mathcal{E}[\mathbf{u}(0)]$ using the multi-step reciprocal prediction error $\mathbb{E}[\mathbf{u}(0)]$, we can develop a critical sampling and adaptive evolution learning algorithm. The central idea is to select samples from locations with large values of error $\mathbb{E}[\mathbf{u}(0)]$ using the following iterative peak finding algorithm. Note that $\mathbf{u}(0) \in \mathbb{R}^n$. Write $\mathbf{u}(0) = [u_1, u_2, \cdots, u_n]$. Let $\mathcal{S}_F^m = \{[\mathbf{u}^j(0) \to \mathbf{u}^j(\Delta)] : 1 \le j \le J_m\}$ be the current sample set. To determine the locations of new samples, $\{\mathbf{u}^j(0) | J_m + 1 \le j \le J_{m+1}\}$, we find the peak value of multi-step reciprocal prediction error $\mathbb{E}[\mathbf{u}(0)]$ at every sampling point $\mathbf{u}(0)$ in the solution space $D$. In our experiment, we choose the sample point $\mathbf{u}(0)$ from the augmented sample set $\bar{\mathcal{S}}_F^n$ defined in (8) of the following section. The corresponding peak location is chosen to be $\mathbf{u}^{J_m+1}(0)$ and the corresponding sample $[\mathbf{u}^{J_m+1}(0) \to \mathbf{u}^{J_m+1}(\Delta)]$ is collected. This process is repeated for $J_{m+1} - J_m$ times to collect $J_{m+1} - J_m$ samples in $\mathbf{\Omega}_m$, which is added to the current sample set:

$$\mathcal{S}_F^{m+1} = \mathcal{S}_F^m \bigcup \mathbf{\Omega}^m = \{[\mathbf{u}^j(0) \to \mathbf{u}^j(\Delta)] : 1 \le j \le J_{m+1}\}. \tag{7}$$

### 3.3 Joint Spatial Dynamics and Temporal Evolution Learning

Joint spatial dynamics and temporal evolution learning aims to couple local dynamics learning in the spatial domain and evolution learning in the temporal domain to achieve robust system evolution learning from the small set of selected critical samples.

**(1) Sample augmentation based on local spatial dynamics.** Let $\mathcal{S}_F^m = \{[\mathbf{u}^j(0) \to \mathbf{u}^j(\Delta)] : 1 \le j \le J_m\}$ be the current set of samples collected from the dynamical system. In this work, $J_m$ is a small number. For example, in our experiments, $J_m$ is in the range of a few hundreds. From our experiments, we find that this learning process is unstable since the number of samples is too small. Our central idea is to design a spatial dynamics network to learn the local spatial dynamics so that we can predict or interpolate more samples at unknown locations from existing samples at known locations. Specifically, let $\{\mathbf{v}^i(0) \in \mathbb{R}^n : 1 \le i \le I\}$ be a large set of randomly selected points in the state space of $\mathbb{R}^n$. Let $\mathbf{v}^i(\Delta) = \mathbf{\Phi}_\Delta(\mathbf{v}^i(0))$ be the system's future state at time $\Delta$ when its current state is $\mathbf{v}^i(0)$. Here, $\{\mathbf{v}^i(\Delta)\}$ are predicted by the spatial dynamics network from the existing sample set $\mathcal{S}_F^m$, denoted by $\hat{\mathbf{v}}^i(\Delta)$. They are added to $\mathcal{S}_F^m$ as augmentation samples

$$\bar{\mathcal{S}}_F^m = \mathcal{S}_F^m \bigcup \mathcal{V}_F, \quad \mathcal{V}_F = \{[\mathbf{v}^i(0) \to \hat{\mathbf{v}}^i(\Delta)] : 1 \le i \le I, \ \hat{\mathbf{v}}^i(\Delta) = \mathbf{\Gamma}_w^m[\mathcal{S}_F^m; \mathbf{v}^i(0)]\}, \tag{8}$$

where $\mathbf{\Gamma}_w^m[\mathcal{S}_F^m; \mathbf{v}^i(0)]$ represent the spatial dynamics network at $m$-th iteration which predicts the future state of $\mathbf{v}^i(0)$ based local spatial change patterns using the existing samples $\mathcal{S}_F^m$. With this augmented sample set $\bar{\mathcal{S}}_F^m$, we can train the temporal evolution network $\mathcal{F}_\theta^m$.

**(2) Learning the local spatial dynamics.** The dynamical system may exhibit highly nonlinear and complex behavior in the whole spatial domain, which could be challenging to be accurately modeled and predicted. However, within a small local neighborhood, its behavior will be much simpler and can be effectively learned by our spatial dynamics network $\mathbf{\Gamma}_w$. Specifically, given an arbitrary point $\mathbf{v}(0) = [v_1, v_2, \cdots, v_n]$ in $\mathbb{R}^n$, we find its nearest $H$ points from the existing sample set $\mathcal{S}_F^m$, and the corresponding samples are denoted by $\mathcal{S}_{\mathbf{v(0)}} = \{[\mathbf{z}^h(0) \to \mathbf{z}^h(\Delta)] : 1 \le h \le H\}$, which are the input to the spatial dynamics network. We use a $p$th-order $n$-variate polynomial $\mathcal{P}(\mathbf{u}) = \mathcal{P}[c_1, c_2, \cdots, c_P](u_1, u_2, \cdots, u_n)$ to locally approximate the local spatial dynamics. The coefficients of the polynomial are $[c_1, c_2, \cdots, c_P]$, which are predicted by the spatial dynamics network $\mathbf{\Gamma}_w$. For example, if $p = 1$, this becomes a linear approximation with $P = n+1$ coefficients. If $p = 2$, the number of coefficients, or the size of the network output becomes $P = \frac{1}{2}(n+2)(n+1)$. To summarize, the task of the spatial dynamics network $\mathbf{\Gamma}_w$ is to predict the coefficients of the polynomial $\mathcal{P}(\mathbf{u})$ from the set of $M$ neighboring samples $\mathcal{S}_{\mathbf{v(0)}}$ so that $\mathcal{P}(\mathbf{u})|_{\mathbf{u}=\mathbf{v}(0)} = \mathbf{v}(\Delta)$, where $\mathbf{v}(\Delta)$ is the future state of the system at time $\Delta$ when its current state is $\mathbf{v}(0)$, or $\mathbf{v}(\Delta) = \mathbf{\Phi}_\Delta(\mathbf{v}(0))$.

When training the spatial dynamics network $\mathbf{\Gamma}_w^m$, we can choose the sample from the existing sample set $\mathcal{S}_F^m$ as the input $\mathbf{v}(0)$ and the corresponding output as $\mathbf{v}(\Delta)$. The $L_2$ loss between the predicted state at time $\Delta$ and its true value for $\mathbf{v}(0)$, namely,

$$\mathbb{L}_{SDN} = \sum_{\mathbf{v}(0) \in \mathcal{S}_F^m} \left\| \mathbf{v}(\Delta) - \mathbf{\Gamma}_w^m[\mathcal{S}_F^m; \mathbf{v}(0)] \right\|^2. \tag{9}$$

**(3) Joint learning of spatial dynamics and temporal evolution networks.** The temporal evolution network $\mathcal{F}_\theta$ and the spatial dynamics network $\mathbf{\Gamma}_w$ aim to characterize the system behavior from two different perspectives, the temporal and spatial domains. In this work, we couple these two networks so that they can learn more effectively. Specifically, we use the spatial dynamics network $\mathbf{\Gamma}_w^m$ to generate a large set of samples $\mathcal{V}_\mathbf{F}$, which is added to the existing samples $\mathcal{S}_F^m$, as explained in (8). This augmented sample set $\bar{\mathcal{S}}_F^m$ is used to train the temporal evolution network $\mathcal{F}_\theta^m$. Note that both networks are predicting the system future state from spatial and temporal domains. Therefore, we can introduce consistency constraint between them. Specifically, let $\mathcal{Q} = \{\mathbf{q}_l \in \mathbb{R}^n : 1 \le l \le L\}$ be a set of randomly generated points in $\mathbb{R}^n$. We use both networks to predict the future state at time $\Delta$ for each $\mathbf{q}_l$ at the initial state. The following consistency loss is used to train both networks:

$$\mathbb{L}_C = \sum_{l=1}^L \left\| \mathcal{F}_\theta^m[\mathbf{q}_l] - \mathbf{\Gamma}_w^m[\mathcal{S}_F^m; \mathbf{q}_l] \right\|^2, \tag{10}$$

## 3.4 THEORETICAL UNDERSTANDING

This section provides some mathematical analysis results to understand and characterize the performance of the proposed critical sampling and adaptive evolution learning method.

**Assumption 3.1.** Assume that the underlying dynamical system $\frac{d}{dt}\mathbf{u}(t) = \mathcal{H}(\mathbf{u}(t))$ is autonomous with $\mathcal{H}$ being Lipschitz continuous (the Lipschitz constant is denoted by $C_\mathcal{H}$) on a set $D \subset \mathbb{R}^n$, which is a bounded region where we are interested in the solution behavior.

**Assumption 3.2.** Suppose that the generalization error of the trained neural network is bounded:

$$\|\mathcal{F}_\theta - \mathbf{\Phi}_\Delta\|_{L^\infty(D)} =: \epsilon_f < +\infty, \quad \|\mathcal{G}_\vartheta - \mathbf{\Psi}_\Delta\|_{L^\infty(D)} =: \epsilon_g < +\infty. \tag{11}$$

We derive a generic bound for the prediction error of our network model. Let $\hat{\mathbf{u}}^{(k)}$ be the forward predicted solution by the trained primal network model at time $t^{(k)} := t_0 + k\Delta$ starting from $t_0$, and $\tilde{\mathbf{u}}^{(k)}$ be the backward predicted solution by the trained dual network model at time $t^{(k)}$ starting from $t^{(K)}$, where $0 \le k \le K$. Denote the corresponding forward prediction error as $\hat{\mathcal{E}}^{(k)} := \|\hat{\mathbf{u}}^{(k)} - \mathbf{u}(t^{(k)})\|$ and the backward prediction error as $\tilde{\mathcal{E}}^{(k)} := \|\tilde{\mathbf{u}}^{(k)} - \mathbf{u}(t^{(k)})\|$, $k = 0, 1, \ldots, K$.

**Theorem 3.3.** *Under Assumptions 3.1 and 3.2, we have the following estimates.*

*(1) If $\hat{\mathbf{u}}^{(k)}, \mathbf{u}(t^{(k)}) \in \hat{D}_\Delta$ for $0 \le k \le K$ with $\hat{D}_\Delta := \{\mathbf{u} \in D : \mathbf{\Phi}_t(\mathbf{u}) \in D \ \forall t \in [0, \Delta]\}$, then*

$$\hat{\mathcal{E}}^{(k)} \le \hat{\mathcal{E}}^{(0)} e^{C_\mathcal{H} k\Delta} + \left( \frac{e^{C_\mathcal{H} k\Delta} - 1}{e^{C_\mathcal{H}\Delta} - 1} \right) \epsilon_f. \tag{12}$$

*(2) If $\tilde{\mathbf{u}}^{(k)}, \mathbf{u}(t^{(k)}) \in \tilde{D}_\Delta$ for $0 \leq k \leq K$ with $\tilde{D}_\Delta := \{\mathbf{u} \in D : \Phi_t^{-1}(\mathbf{u}) \in D \ \forall t \in [0, \Delta]\}$, then*

$$\tilde{\mathcal{E}}^{(k)} \leq \tilde{\mathcal{E}}^{(K)} e^{C_{\mathcal{H}}(K-k)\Delta} + \left( \frac{e^{C_{\mathcal{H}}(K-k)\Delta} - 1}{e^{C_{\mathcal{H}}\Delta} - 1} \right) \epsilon_g. \tag{13}$$

Please see Section C for the detailed proof of this theorem and further discussions in Theorem C.4. The estimates (12)–(13) imply that in the worst cases the prediction errors may grow exponentially with the number of steps $k$. The analysis suggests that the reciprocal prediction error is correlated with the network modeling error, which provides a theoretical support for our finding. The critical sampling can help to effectively reduce the reciprocal prediction error and suppress the undesirable error growth, thereby enhancing the accuracy and stability of our model.

## 4 EXPERIMENTAL RESULTS

### 4.1 EXPERIMENTAL SETTINGS

We follow the evaluation procedure used in existing research, for example those reviewed in Section 2, to evaluate the performance of our proposed method on specific examples of dynamical systems. We consider four representative systems with ODEs and PDEs as their governing equations, as summarized in Table 2 in Supplemental Materials. They include (1) the Damped Pendulum ODE equations in $\mathbb{R}^2$, (2) a nonlinear ODE equation in $\mathbb{R}^2$, (3) the Lorenz system (ODE) in $\mathbb{R}^3$, and (4) the Viscous Burgers' equation (PDE). Note that, for the final PDE system, we approximate it in a generalized Fourier space to reduce the problem to finite dimensions as in Wu & Xiu (2020). We use the projection operator $\mathcal{P}_n : \mathbb{V} \to \mathbb{V}_n$, where $\mathbb{V}_n = \text{span}\{\sin(jx) : 1 \leqslant j \leqslant n\}$ with $n = 9$. Certainly, our proposed method can be also applied to many other dynamical systems, we simply use these four example systems to demonstrate the performance of our new method. In Section A.2, we provide detailed descriptions on how to obtain the training samples for the dynamical systems. An overview and pseudo-code of our method can be found in Section A.4.

### 4.2 PERFORMANCE RESULTS

We choose the evolution learning method developed in Qin et al. (2019); Wu & Xiu (2020) as our baseline. This method has achieved impressive performance in learning the evolution behaviors of autonomous systems and attracted much attention from the research community. On top of this method, we implement our proposed method of critical sampling and adaptive evolution learning. We demonstrate that, to achieve the same modeling error, our method needs much fewer samples.

Table 1 compares the numbers of samples needed for learning the system evolution by the baseline method and our critical sampling and adaptive learning method. For example, for the Lorenz system, it needs 1000000 samples to achieve the modeling error of 0.197. Using our proposed critical sampling method, the number of samples can be reduced to 1765, while achieving an even smaller modeling error 0.194. The number of samples has been reduced by 567 times.

Table 1: Samples for learning the system evolution using the baseline method and our method. The prediction errors are evaluated on 50 arbitrarily chosen solution trajectories in the computational domain. All testing states are not included in the training set. Average error and standard deviation are reported for each dynamical system. See Section A.3 in Supplemental Materials for more details.

| Dynamical System | Baseline | | Our Work | | Ratio |
|---|---|---|---|---|---|
| | Samples | Prediction Error | Samples | Prediction Error | |
| Damped Pendulum | 14400 | $0.02630 \pm 0.01200$ | **417** | **0.02411** $\pm 0.00991$ | 34.53 |
| 2D Nonlinear | 14400 | $0.00037 \pm 0.00021$ | **925** | **0.00035** $\pm 0.00015$ | 15.57 |
| Lorenz System | 1000000 | $0.19685 \pm 0.07768$ | **1765** | **0.19357** $\pm 0.05695$ | 566.57 |
| Viscous Burgers' Eq. | 500000 | $0.01679 \pm 0.00878$ | **19683** | **0.01652** $\pm 0.00818$ | 25.40 |

Figure 5 shows the performance comparison results for the four dynamical systems listed in Table 2. In each sub-figure, the horizontal dashed line shows the average network modeling error achieved by the baseline method for the number of samples shows in the legend. This number is empirically chosen since it is needed for the network to achieve a reasonably accurate and robust learning performance. We can see that as more and more samples are selected by our critical sampling method, the network modeling error quickly drops below the average modeling error of the baseline method.

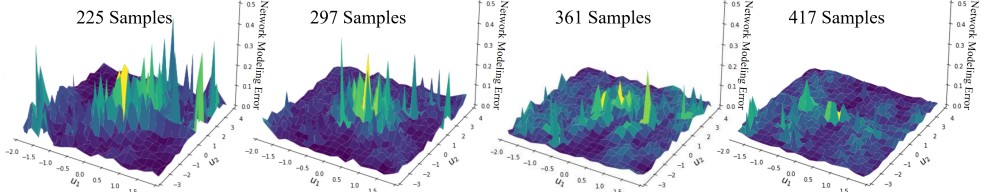

Figure 5: The critical sampling and adaptive learning results on four dynamical systems.

Figure 6 shows that the network modeling error $\mathcal{E}[\mathbf{u}(0)]$ of the Damped Pendulum system, is being quickly reduced with more and more critical samples are collected. In Figure 7, the top row shows the phase portraits of the solutions for two systems, Damped Pendulum (left) and 2D Nonlinear system (right), obtained by our method with comparison against the ground-truth solutions. The bottom row shows an example solution for the Viscous Burgers' PDE system. The first one is the ground-truth solution. The second and third figures are the solution and its difference from the ground-truth solution for the baseline method. The last two figures are the solution and difference obtained by our method. We can see that using fewer samples, our method is able to learn the system evolution and predict its future states at the same level of accuracy. In Section D, we provide more experimental results to further understand the performance of our proposed method.

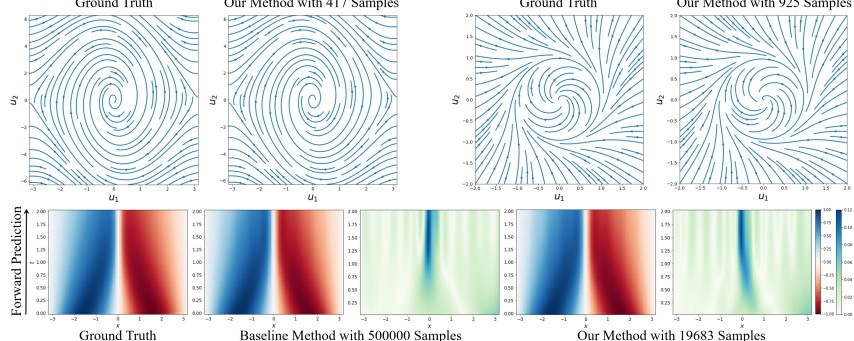

Figure 6: The reduction of network modeling error with more and more critical samples are collected for the Damped Pendulum system.

Figure 7: Solutions obtained by critical sampling and adaptive evolution learning method.

# 5 CONCLUSION AND FURTHER DISCUSSION

In this work, we have studied the critical sampling for adaptive evolution operator learning problem. We have made an interesting finding that the network modeling error is correlated with the multi-step reciprocal prediction error. With this, we are able to perform a dynamic selection of critical samples from regions with high network modeling errors and develop an adaptive sampling-learning method for dynamical systems based on the spatial-temporal evolution network. Extensive experimental results demonstrate that our method is able to dramatically reduce the number of samples needed for effective learning and accurate prediction of the evolution behaviors.

In the future, we hope to apply our approach to large-scale dynamical systems, by combining some reduced-order modeling or lifting techniques (cf. Qian et al. (2020)) or incorporating certain sparsity (cf. Schaeffer et al. (2018)). Another important question that has not been fully addressed in this paper is how to control the system state towards those samples selected by our critical sampling method. During simulations, this system state control is often available. However, for some complex systems, the exact change of the system state may not be trivial. In this case, we shall investigate how the system state control impacts the critical sampling and system modeling performance.

## REPRODUCIBILITY STATEMENT

We provide detailed data generation process and experiment settings in Section A of Supplemental Materials. An overview and pseudo-code of our method can be found in Section A.4 of Supplemental Materials.

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

# SUPPLEMENTAL MATERIALS

In Supplemental Materials, we provide more algorithm implementation details, further discussion on related work, further analytical understanding, and additional experimental results for in-depth understanding of our proposed method.

## A  EXPERIMENTAL SETTINGS AND IMPLEMENTATION DETAILS

In this section, we provide more details on system configurations and algorithm implementation.

### A.1  DYNAMICAL SYSTEMS STUDIED IN THIS WORK

In this paper, we consider four representative systems with ODEs and PDEs as their governing equations. They include (1) the Damped Pendulum ODE in $\mathbb{R}^2$, (2) a nonlinear ODE system in $\mathbb{R}^2$, (3) the Lorenz system (ODE) in $\mathbb{R}^3$, and (4) the Viscous Burgers' equation (PDE). Their governing equations are shown in Table 2.

Table 2: Overview of the 4 governing equation systems we demonstrate in this work.

| SYSTEM | GOVERNING EQUATIONS |
|---|---|
| DAMPED PENDULUM EQUATION | $\begin{cases} \frac{d}{dt}u_1 = u_2, \\ \frac{d}{dt}u_2 = -0.2u_2 - 8.91\sin u_1. \end{cases}$ |
| A 2D NONLINEAR EQUATION | $\begin{cases} \frac{d}{dt}u_1 = u_2 - u_1\left({u_1}^2 + {u_2}^2 - 1\right), \\ \frac{d}{dt}u_2 = -u_1 - u_2\left({u_1}^2 + {u_2}^2 - 1\right). \end{cases}$ |
| LORENZ SYSTEM | $\begin{cases} \frac{d}{dt}u_1 = 10\left(u_2 - u_1\right), \\ \frac{d}{dt}u_2 = u_1\left(28 - u_3\right) - u_2, \\ \frac{d}{dt}u_3 = u_1u_2 - (8/3)u_3. \end{cases}$ |
| VISCOUS BURGERS' EQUATION | $\begin{cases} u_t + \left(\frac{u^2}{2}\right)_x = 0.1u_{xx}, \quad (x,t) \in (-\pi, \pi) \times \mathbb{R}^+, \\ u(-\pi, t) = u(\pi, t) = 0, \quad t \in \mathbb{R}^+. \end{cases}$ |

### A.2  SYSTEM CONFIGURATIONS

For the ODE examples, we follow the procedure in Qin et al. (2019) to generate the training data pairs $\{[\mathbf{u}^j(0), \mathbf{u}^j(\Delta)]\}$ as follows. First, we generate $J$ system state vectors $\{\mathbf{u}^j(0)\}_{j=1}^J$ at time 0 based on a uniform distribution over a computational domain $D$. Here, $D$ is the region where we are interested in the solution space. It is typically chosen to be a hypercube prior to the computation, which will be explained in the following. Then, for each $j$, starting from $\mathbf{u}^j(0)$, we solve the true ODEs for a time lag of $\Delta$ using a highly accurate ODE solver to generate $\mathbf{u}^j(\Delta)$. *Notice that, once the data is generated, we assume that the true equations are unknown, and the sampled data pairs are the only known information during the learning process.*

For the first example dynamical system listed in Table 2, its computational domain is $D = [-\pi, \pi] \times [-2\pi, 2\pi]$. We choose $\Delta = 0.1$. For the second system, the computational domain is $D = [-2, 2]^2$. The time lag $\Delta$ is set as $0.1$. For the third system, the computational domain is $D = [-25, 25]^2 \times [0, 50]$. The time lag $\Delta$ is set as $0.01$.

For the Viscous Burgers' PDE system, because the evolution operator is defined between infinite dimensional spaces, and we approximate it in a modal space, namely, a generalized Fourier space, in order to reduce the problem to finite dimensions as in Wu & Xiu (2020). We follow the same

procedure specified in Wu & Xiu (2020) to choose a basis of the finite dimensional space $\mathbb{V}_n$ to represent the solutions, then apply the projection operator to project the snapshot data to $\mathbb{V}_n$ to obtain the training data in the generalized Fourier space. The choice of basis functions is fairly flexible, any basis suitable for spatial approximation of the solution data can be used. Once the basis functions are selected, a projection operator $\mathcal{P}_n : \mathbb{V} \rightarrow \mathbb{V}_n$ is applied to obtain the solution in the finite dimensional form.

The approximation space is chosen to be relatively larger as $\mathbb{V}_n = \text{span} \left\{ \sin(jx) : 1 \leqslant j \leqslant n \right\}$ with $n = 9$. The time lag $\Delta$ is taken as 0.05. The domain $D$ in the modal space is set as $[-1.5, 1.5] \times [-0.5, 0.5] \times [-0.2, 0.2]^2 \times [-0.1, 0.1]^2 \times [-0.05, 0.05]^2 \times [-0.02, 0.02]$, from which we sample the training samples.

Our task is to demonstrate how our proposed method is able to significantly reduce the number of samples needed for evolution learning. Specifically, for the baseline method, we use random sampling, randomly selecting locations in the solution space to collect samples for evolution learning. For example, for the first dynamical system, Damped Pendulum system (ODE) in a 2-D space, the baseline method use 14400 samples to achieve an average network modeling error of 0.026. We then use our method to adaptively discover critical samples and refine the evolution network to reach the same or even smaller network modeling error. We demonstrate that, to achieve the same modeling error, our proposed method needs much fewer samples.

## A.3 IMPLEMENTATION DETAILS

In all examples, we use the recursive ResNet (RS-ResNet) architecture in He et al. (2016); Qin et al. (2019), which is a block variant of the ResNet and has been proven in Qin et al. (2019); Wu & Xiu (2020) to be highly suitable for learning flow maps and evolution operators.

For all the 4 systems, the batch size is set as 10. In the two 2-dimensional ODE systems, we use the one-block ResNet method with each block containing 3 hidden layers of equal width of 20 neurons, while in the 3-dimensional ODE system, we use the one-block ResNet method with each block containing 3 hidden layers of equal width of 30 neurons. For the final PDE system, we use the four-block ResNet method with each block containing 3 hidden layers of equal width of 20 neurons. Adam optimizer with betas equal $(0.9, 0.99)$ is used for training. In the two 2-dimensional ODE systems, all the networks are trained with 150 epochs. In the Lorenz system and Viscous Burgers' equation, all the networks are trained with 60 epochs. The initial learning rate is set as $10^{-3}$ , and will decay gradually to $10^{-6}$ during the training process. All networks are trained using PyTorch on one RTX 3060 GPU.

In the four example systems, we evaluate the performance of our models on time duration $t \in [0, 20]$, $t \in [0, 10]$, $t \in [0, 5]$, $t \in [0, 2]$, respectively. For the first two ODE systems, the network modeling error is evaluated by average MSE error at each time step on 50 different arbitrarily chosen solution trajectories. For the Lorenz system, we evaluate the network by average MSE error at each time step under 50 different initial conditions. For the final PDE system, the network modeling error is evaluated by the average $L_2$ norm error on 100 points at time $t = 2$ under 50 different initial conditions.

## A.4 PSEUDO-CODE AND OVERVIEW OF OUR PROPOSED METHOD

Our proposed method of critical sampling and adaptive evolution operator learning algorithm is summarized in Algorithm 1. Our proposed method has the following steps: First, we generate $J_m$ data pairs based on a uniform distribution over a computational domain $D$ using highly accurate ODE/PDE solver. Then, we train the spatial dynamics network $\mathbf{\Gamma}_w^m$ using the generated data pairs and use the network to generate a large set of additional samples. After that, we train forward evolution network $\mathcal{F}_\theta^m$ using the data pairs, and train backward evolution network $\mathcal{G}_\vartheta^m$ using the reversed data pairs. Multi-step reciprocal prediction error is evaluated on different locations in computational domain $D$ with both networks. Finally, we collect the samples from the location with peak reciprocal prediction error. Those samples should be added to the initial set and the whole process is repeated until the network modeling error $\mathcal{E}[\mathbf{u}(0)]$ is smaller than threshold.

---

**Algorithm 1:** Critical Sampling and Adaptive Evolution Operator Learning Algorithm

---

**Input:** Number of samples in the initial set $J_m$; Number of samples in the updated set $J_{m+1}$; Batch size; Learning rate; Adam hyperparameters; Initial parameters of networks.

**Output:** Optimized forward evolution network $\mathcal{F}_\theta$.

1 Generate $J_m$ data pairs based on a uniform distribution over a computational domain $D$, initialize the sample set $\mathcal{S}_F^m = \{[\mathbf{u}^j(0) \to \mathbf{u}^j(\Delta)] : 1 \le j \le J_m\}$;

2 **repeat**

   // Sample augmentation based on local spatial dynamics.

3    Train spatial dynamics network $\mathbf{\Gamma}_w^m$ using $\mathcal{S}_F^m$;

4    Use $\mathbf{\Gamma}_w$ to generate a large set of samples $\mathcal{V}_F$, add to the existing sample set:
   $\bar{\mathcal{S}}_F^m = \mathcal{S}_F^m \bigcup \mathcal{V}_F$;

   // Multi-step reciprocal prediction.

5    Reverse the data pairs in $\bar{\mathcal{S}}_F^m$ to get $\bar{\mathcal{S}}_G^m$;

6    Train forward evolution network $\mathcal{F}_\theta^m$ using $\bar{\mathcal{S}}_F^m$, backward evolution network $\mathcal{G}_\vartheta^m$ using $\bar{\mathcal{S}}_G^m$;

7    Perform $K$-step forward prediction using $\mathcal{F}_\theta^m$ to get $\hat{\mathbf{u}}(k\Delta)$;

8    Perform $K$-step backward prediction using $\mathcal{G}_\vartheta^m$ to get $\bar{\mathbf{u}}(k\Delta)$;

9    Calculate multi-step reciprocal prediction error $\mathbb{E}[\mathbf{u}(0)] = \sum_{k=0}^{K} \left\| \hat{\mathbf{u}}(k\Delta) - \bar{\mathbf{u}}(k\Delta) \right\|^2$;

   // Critical sampling.

10    Initialize critical sample set $\mathbf{\Omega}_m$ to empty set;

11    **for** $i$ **in** $\{1, 2, \dots, J_{m+1} - J_m\}$ **do**

12      Choose location with peak reciprocal prediction error $\mathbb{E}[\mathbf{u}(0)]$ to be $\mathbf{u}^{J_m+i}(0)$;

13      Collect corresponding sample $[\mathbf{u}^{J_m+i}(0) \to \mathbf{u}^{J_m+i}(\Delta)]$, add to $\mathbf{\Omega}_m$.

14    **end**

15    Add the critical sample set to the current sample set (without augmented samples $\mathcal{V}_F$):
   $\mathcal{S}_F^{m+1} = \mathcal{S}_F^m \bigcup \mathbf{\Omega}_m$;

16    $\mathcal{S}_F^m = \mathcal{S}_F^{m+1}$, $J_m = J_{m+1}$;

17 **until** *network modeling error $\mathcal{E}[\mathbf{u}(0)]$ smaller than threshold*;

---

# B   FURTHER DISCUSSION ON RELATED WORK AND OUR UNIQUE CONTRIBUTIONS

For the data-driven discovery of ODEs and PDEs, there are two major popular approaches. The first approach typically tries to identify all the potential terms in the unknown governing equations from a priori dictionary, which includes all possible terms that may appear in the equations (Brunton et al., 2016; Schaeffer & McCalla, 2017; Rudy et al., 2017; Raissi, 2018; Long et al., 2018; Wu & Xiu, 2019; Wu et al., 2020; Xu & Zhang, 2021). The second approach for data-driven learning of unknown dynamical systems is to approximate the evolution operator of the underlying equations, instead of identifying the terms in the equations (Qin et al., 2019; Wu & Xiu, 2020; Qin et al., 2021a; Chen et al., 2022; Li et al., 2021b).

In this work, we choose the second approach of learning the evolution operator. Compared to the first approach which tries to recover the mathematical expression of the unknown governing equations, the second approach of learning evolution operators often has the following distinctive features: (1) It does *not* require *a prior* knowledge about the form of the unknown governing equations. (2) Unlike the first approach, the evolution operator approach, which is based on the integral from of the underlying dynamical system, does not require numerical approximation of the temporal derivatives and allows large time steps during the learning and prediction process. (3) Although the first approach may successfully recover the expressions of the governing equations, during the prediction stage, it still needs to construct suitable numerical schemes to further solve the learned equations. On the contrary, the second approach learns the evolution operator which can be directly used to perform long-term prediction of the system behavior in the future.

Learning a deep neural network model for the evolution operator often requires a very large number of training (often in the ranges of tens or hundreds of thousands), especially for high-dimensional dynamical systems. Existing methods in the literature often assume that samples are always available. In practice, this is not the case. In field conditions of dynamical systems, sample collection is often costly  or very limited due to resource constraints or experimental accessibility. To our best knowledge, this work is one of the first efforts to address this challenge.

For bandwidth-limited signals, the Nyquist sampling theorem can be used to determine the minimum number of samples needed to perfectly reconstruct the original signals. For sparse signals, compressive sensing theorems have been developed in the past years to determine the number of observations needed to reconstruct the original signals. However, for dynamical systems with highly nonlinear and dynamic behaviors, it is very challenging to develop corresponding analytical results for minimum sampling. This task becomes even more challenging for systems whose governing laws or equations are unknown to us.

In this work, instead of pursuing an analytical approach to answer the above critical sampling question, we propose to explore a design-based approach which aims to address the following issues: (1) how to design a deep neural network for dynamical systems which robustly learn its evolution behavior with a smaller set of samples and effectively improve its prediction accuracy with more and more critical samples being added? (2) How to design an algorithm to dynamically determine the most important samples or critical samples for network model learning? (3) How can we couple the deep neural network model learning process with the critical sampling algorithm to establish an adaptive joint sampling-learning method for dynamical systems to drastically reduce the number of needed samples for effective learning of the system evolution behavior?

Although our paper and Qin et al. (2019); Wu & Xiu (2020) study the same topic "learning the evolution behavior of unknown dynamical systems", our paper focuses on investigating a **different yet very important** problem: how to learn the evolution behavior of unknown dynamic systems with very few samples and how to dynamically select these samples. This paper has significant impacts in practice since collecting samples from real-world dynamical systems can be very costly or very limited due to resource/labor constraints or experimental accessibility.

The success of our approach also contributes significantly to deep learning and signal estimation research. It investigates an important problem in learning and prediction with deep neural networks: *how to characterize and estimate the prediction errors during the inference stage?* This is a very challenging problem since the ground-truth values for unknown dynamical systems are not available.

This problem was not studied in Qin et al. (2019); Wu & Xiu (2020) which present a ResNet approach to model the evolution behavior of the dynamical systems. Those methods typically assumed availability of sufficient data, and often used tens of thousands or even millions of samples for training.

When addressing these important problems, our paper has the following unique novelties and contributions: **First**, we have introduced the concept of multi-step reciprocal prediction error based on a prediction loop of forward and backward deep neural works. When modeling the evolution behavior of unknown dynamical systems, we discovered that the prediction error of the deep neural networks is highly correlated with the multi-step reciprocal prediction error. **Second**, most interestingly, unlike existing methods in prediction error estimation which often replies on assumptions about the underlying data statistics or distributions. Our method computes the prediction error based on the single input data itself. **Third**, we have introduced the approach of spatial-temporal evolution network learning approach which couples temporal evolution learning with spatial dynamics learning to significantly improve the learning performance with very few samples.

**Relations to and Differences from Adjoint State Method.** The adjoint state method (Pontryagin, 1962) is a numerical method for efficiently computing the gradients of functions for numerical optimization problems. Recently, it has been successfully used for highly efficient learning the ODE for dynamical systems (e.g., (Chen et al., 2018; Zhuang et al., 2020b;a)). It has also been further extended using the adaptive check point method for more accurate estimation of the gradients for neural ODE (Chen et al., 2018). To address the memory cost and the inaccuracy in reverse-time trajectory, a memory-efficient ALF Integrator (MALI) has been developed (Zhuang et al., 2020a). Our multi-step forward and backward processes and the usage of mismatch errors to adjust the estimation of our proposed method look similar to those in the recent adjoint state methods (Chen et al.,

2018; Zhuang et al., 2020a;b). However, our method is uniquely different in the following aspects. (1) In the adjoint state method, the backward propagation is performed for computing the gradients of loss function. However, in our method, the backward network is for learning the inverse operator of the forward evolution operator (namely, the backward evolution operator, please see Lemma C.1). (2) The adjoint state method aims to compute the gradients of the loss functions more efficiently and more accurately. However, our method aims to discover the critical samples for network learning. The purpose of our multi-step reciprocal error using the forward and backward networks is to estimate the network prediction error. We have made a very interesting finding: this multi-step reciprocal error is highly correlated with network prediction error. Note that this prediction error cannot be obtained since the ground-truth is not available. This finding is very important in our network learning and prediction. It also allows us to perform critical sample discovery for unknown dynamical systems and learn the system behavior with very few samples. Once the critical samples are discovered, the network is trained as usual. No adjoint state methods are used here to compute the gradients during the training process.

## C  Further Analytical Understanding of Our Proposed Method

In this section, we provide further analytical understanding of our proposed method of critical sampling and adaptive evolution learning, especially on the error bound of evolution operator learning. Let us consider the following autonomous ODE system as an example:

$$\frac{d\mathbf{u}(t)}{dt} = \mathcal{H}(\mathbf{u}(t)), \quad t \in \mathbb{R}^+, \tag{14}$$

where $\mathbf{u}(t) \in \mathbb{R}^n$ are the state variables. Let $\mathbf{\Phi}_\Delta : \mathbb{R}^n \to \mathbb{R}^n$ be the *evolution operator*, which maps the system state from time $t = 0$ to its next state at time $\Delta$: $\mathbf{u}(t + \Delta) = \mathbf{\Phi}_\Delta(\mathbf{u}(t))$. It should be noted that, for autonomous systems, this evolution operator $\mathbf{\Phi}_\Delta$ remains invariant for different time instance $t$. It only depends on the time difference $\Delta$.

**Lemma C.1.** *The backward evolution operator $\Psi_\Delta$ of system (14) is actually the forward evolution operator of the following dynamical system*

$$\frac{d}{dt}\bar{\mathbf{u}}(t) = -\mathcal{H}(\bar{\mathbf{u}}(t)). \tag{15}$$

*Proof.* Define $\bar{\mathbf{u}}(t) := \mathbf{u}(T - t)$ for an arbitrarily fixed $T > 0$. It can be seen that

$$\frac{d}{dt}\bar{\mathbf{u}}(t) = \frac{d}{dt}(\mathbf{u}(T - t)) = -\left.\frac{d\mathbf{u}}{dt}\right|_{T-t} = -\mathcal{H}(\mathbf{u}(T - t)) = -\mathcal{H}(\bar{\mathbf{u}}(t)).$$

This means $\bar{\mathbf{u}}(t)$ satisfies the ODEs (15). The forward evolution operator of system (15), which maps $\bar{\mathbf{u}}(t)$ to $\bar{\mathbf{u}}(t + \Delta)$, is equivalent to the mapping from $\mathbf{u}(T - t)$ to $\mathbf{u}(T - t - \Delta)$, which exactly coincides with the backward evolution operator $\Psi_\Delta$ of system (14). The proof is completed. $\square$

In the following analysis, we will always use Assumption 3.1 with $\mathcal{H}$ being Lipschitz continuous (the Lipschitz constant is denoted by $C_\mathcal{H}$) on a set $D \subset \mathbb{R}^n$. Here $D$ is a bounded region where we are interested in the solution behavior.

**Lemma C.2.** *Suppose Assumptions 3.1 holds. Define*

$$\hat{D}_\Delta := \{\mathbf{u} \in D : \Phi_t(\mathbf{u}) \in D \ \forall t \in [0, \Delta]\}.$$

*The forward evolution operator $\Phi_\Delta$ of system (14) is Lipschitz continuous on $\hat{D}_\Delta$, i.e., for any $\mathbf{u}_1, \mathbf{u}_2 \in \hat{D}_\Delta$,*

$$\|\Phi_\Delta(\mathbf{u}_1) - \Phi_\Delta(\mathbf{u}_2)\| \leq e^{C_\mathcal{H}\Delta}\|\mathbf{u}_1 - \mathbf{u}_2\|. \tag{16}$$

*Proof.* This follows from a classical result in the dynamical system; see (Stuart & Humphries, 1998). $\square$

**Lemma C.3.** *Suppose Assumptions 3.1 holds. Define*

$$\tilde{D}_\Delta := \{\mathbf{u} \in D : \Psi_t(\mathbf{u}) \in D \ \forall t \in [0, \Delta]\}.$$

*The backward evolution operator $\Psi_\Delta$ of system (14) is Lipschitz continuous on $\tilde{D}_\Delta$, i.e., for any $\mathbf{u}_1, \mathbf{u}_2 \in \tilde{D}_\Delta$,*

$$\|\Psi_\Delta(\mathbf{u}_1) - \Psi_\Delta(\mathbf{u}_2)\| \le e^{C_\mathcal{H}\Delta}\|\mathbf{u}_1 - \mathbf{u}_2\|. \tag{17}$$

*Proof.* According to Lemma C.1, $\Psi_\Delta$ is the forward evolution operator of system (15). Following the idea of Lemma C.2 for $\Psi_\Delta$ one can complete the proof. $\square$

We now derive a simple generic bound for the prediction error of our network model. Suppose the generalization error of the trained neural network is bounded: Let $\hat{\mathbf{u}}^{(k)}$ be the forward predicted solution by the trained primal network model at time $t^{(k)} := t_0 + k\Delta$ starting from $t_0$, and $\tilde{\mathbf{u}}^{(k)}$ be the backward predicted solution by the trained dual network model at time $t^{(k)}$ starting from $t^{(K)}$, where $0 \le k \le K$. Denote the corresponding forward prediction error as $\hat{\mathcal{E}}^{(k)} := \|\hat{\mathbf{u}}^{(k)} - \mathbf{u}(t^{(k)})\|$ and the backward prediction error as $\tilde{\mathcal{E}}^{(k)} := \|\tilde{\mathbf{u}}^{(k)} - \mathbf{u}(t^{(k)})\|$, $k = 0, 1, \ldots, K$. We then have the following estimates.

**Theorem C.4.** *Under Assumptions 3.1 and 3.2, we have:*

1. *If $\hat{\mathbf{u}}^{(k)}, \mathbf{u}(t^{(k)}) \in \hat{D}_\Delta$ for $0 \le k \le K - 1$, then*

$$\hat{\mathcal{E}}^{(k)} \le \hat{\mathcal{E}}^{(0)}e^{C_\mathcal{H}k\Delta} + \left(\frac{e^{C_\mathcal{H}k\Delta} - 1}{e^{C_\mathcal{H}\Delta} - 1}\right)\epsilon_f. \tag{18}$$

2. *If $\tilde{\mathbf{u}}^{(k)}, \mathbf{u}(t^{(k)}) \in \tilde{D}_\Delta$ for $0 \le k \le K - 1$, then*

$$\tilde{\mathcal{E}}^{(k)} \le \tilde{\mathcal{E}}^{(K)}e^{C_\mathcal{H}(K-k)\Delta} + \left(\frac{e^{C_\mathcal{H}(K-k)\Delta} - 1}{e^{C_\mathcal{H}\Delta} - 1}\right)\epsilon_g. \tag{19}$$

3. *Furthermore, if we take $\tilde{\mathbf{u}}^{(0)} = \mathbf{u}(t_0)$ and pass $\tilde{\mathbf{u}}^{(K)}$ as the input of the trained dual network with $\tilde{\mathbf{u}}^{(K)} = \hat{\mathbf{u}}^{(K)}$, then*

$$\tilde{\mathcal{E}}^{(k)} \le \min\left\{ \left\|\mathcal{G}_\vartheta^{(K-k)} \circ \mathcal{F}_\theta^{(K-k)} - I\right\|_{L^\infty(D)} + \left(\frac{e^{C_\mathcal{H}k\Delta} - 1}{e^{C_\mathcal{H}\Delta} - 1}\right)\epsilon_f, \right.$$
$$\left. \left(\frac{e^{C_\mathcal{H}K\Delta} - 1}{e^{C_\mathcal{H}\Delta} - 1}\right)\epsilon_f e^{C_\mathcal{H}(K-k)\Delta} + \left(\frac{e^{C_\mathcal{H}(K-k)\Delta} - 1}{e^{C_\mathcal{H}\Delta} - 1}\right)\epsilon_g \right\}, \tag{20}$$

*and in particular when $k = 0$,*

$$\|\tilde{\mathbf{u}}^{(0)} - \mathbf{u}(t_0)\|_2$$
$$\le \min\left\{ \left\|\mathcal{G}_\vartheta^{(K)} \circ \mathcal{F}_\theta^{(K)} - I\right\|_{L^\infty(D)}, \left(\frac{e^{C_\mathcal{H}K\Delta} - 1}{e^{C_\mathcal{H}\Delta} - 1}\right)\left(\epsilon_f e^{C_\mathcal{H}K\Delta} + \epsilon_g\right) \right\}, \tag{21}$$

*where $I$ denotes the identity map.*

*Proof.* Recall that $\hat{\mathbf{u}}^{(k)} = \mathcal{F}_\theta(\hat{\mathbf{u}}^{(k-1)})$ and $\mathbf{u}(t^{(k)}) = \Phi_\Delta(\mathbf{u}(t^{(k-1)}))$. Using the triangle inequality for the Euclidean norm, the assumption (11), and the Lipschitz continuity (16) of $\Phi_\Delta$ in Lemma C.2, we can derive that

$$\hat{\mathcal{E}}^{(k)} = \left\|\hat{\mathbf{u}}^{(k)} - \mathbf{u}(t^{(k)})\right\|$$
$$= \left\|\mathcal{F}_\theta(\hat{\mathbf{u}}^{(k-1)}) - \Phi_\Delta(\mathbf{u}(t^{(k-1)}))\right\|$$
$$\le \left\|\mathcal{F}_\theta(\hat{\mathbf{u}}^{(k-1)}) - \Phi_\Delta(\hat{\mathbf{u}}^{(k-1)})\right\| + \left\|\Phi_\Delta(\hat{\mathbf{u}}^{(k-1)}) - \Phi_\Delta(\mathbf{u}(t^{(k-1)}))\right\|$$
$$\le \|\mathcal{F}_\theta - \Phi_\Delta\|_{L^\infty(D)} + e^{C_\mathcal{H}\Delta}\left\|\hat{\mathbf{u}}^{(k-1)} - \mathbf{u}(t^{(k-1)})\right\|$$
$$= e^{C_\mathcal{H}\Delta}\hat{\mathcal{E}}^{(k-1)} + \epsilon_f.$$

Repeatedly utilizing such an estimate leads to

$$
\begin{aligned}
\hat{\mathcal{E}}^{(k)} &\leq e^{C_{\mathcal{H}}\Delta}\hat{\mathcal{E}}^{(k-1)} + \epsilon_f \\
&\leq e^{2C_{\mathcal{H}}\Delta}\hat{\mathcal{E}}^{(k-2)} + e^{C_{\mathcal{H}}\Delta}\epsilon_f + \epsilon_f \\
&\leq e^{3C_{\mathcal{H}}\Delta}\hat{\mathcal{E}}^{(k-3)} + e^{2C_{\mathcal{H}}\Delta}\epsilon_f^2 + e^{C_{\mathcal{H}}\Delta}\epsilon_f + \epsilon_f \\
&\leq \cdots \\
&\leq e^{kC_{\mathcal{H}}\Delta}\hat{\mathcal{E}}^{(0)} + \epsilon_f \sum_{j=1}^{k-1} e^{jC_{\mathcal{H}}\Delta},
\end{aligned}
$$

which completes the proof of (18). Similarly, for the backward prediction procedure, we recall that $\tilde{\mathbf{u}}^{(k)} = \mathcal{G}_\vartheta(\tilde{\mathbf{u}}^{(k+1)})$ and $\mathbf{u}(t^{(k)}) = \Psi_\Delta(\mathbf{u}(t^{(k+1)}))$, and then use Lemma C.3 to deduce that

$$
\begin{aligned}
\tilde{\mathcal{E}}^{(k)} &= \left\|\tilde{\mathbf{u}}^{(k)} - \mathbf{u}(t^{(k)})\right\| \\
&= \left\|\mathcal{G}_\vartheta(\tilde{\mathbf{u}}^{(k+1)}) - \Psi_\Delta(\mathbf{u}(t^{(k+1)}))\right\| \\
&\leq \left\|\mathcal{G}_\vartheta(\tilde{\mathbf{u}}^{(k+1)}) - \Psi_\Delta(\tilde{\mathbf{u}}^{(k+1)})\right\| + \left\|\Psi_\Delta(\tilde{\mathbf{u}}^{(k+1)}) - \Psi_\Delta(\mathbf{u}(t^{(k+1)}))\right\| \\
&\leq \|\mathcal{G}_\vartheta - \Psi_\Delta\|_{L^\infty(D)} + e^{C_{\mathcal{H}}\Delta}\left\|\tilde{\mathbf{u}}^{(k+1)} - \mathbf{u}(t^{(k+1)})\right\| \\
&= e^{C_{\mathcal{H}}\Delta}\tilde{\mathcal{E}}^{(k+1)} + \epsilon_g.
\end{aligned}
$$

Repeatedly utilizing such an estimate leads to

$$
\begin{aligned}
\tilde{\mathcal{E}}^{(k)} &\leq e^{C_{\mathcal{H}}\Delta}\tilde{\mathcal{E}}^{(k+1)} + \epsilon_g \\
&\leq e^{2C_{\mathcal{H}}\Delta}\tilde{\mathcal{E}}^{(k+2)} + e^{C_{\mathcal{H}}\Delta}\epsilon_g + \epsilon_g \\
&\leq e^{3C_{\mathcal{H}}\Delta}\tilde{\mathcal{E}}^{(k+3)} + e^{2C_{\mathcal{H}}\Delta}\epsilon_g^2 + e^{C_{\mathcal{H}}\Delta}\epsilon_g + \epsilon_g \\
&\leq \cdots \\
&\leq e^{(K-k)C_{\mathcal{H}}\Delta}\tilde{\mathcal{E}}^{(K)} + \epsilon_g \sum_{j=1}^{K-k-1} e^{jC_{\mathcal{H}}\Delta},
\end{aligned}
$$

which completes the proof of (19). Furthermore, if we take $\tilde{\mathbf{u}}^{(0)} = \mathbf{u}(t_0)$ and pass $\tilde{\mathbf{u}}^{(K)}$ as the input of the trained dual network with $\tilde{\mathbf{u}}^{(K)} = \hat{\mathbf{u}}^{(K)}$, then $\hat{\mathcal{E}}^{(0)} = 0$ and $\tilde{\mathcal{E}}^{(K)} = \hat{\mathcal{E}}^{(K)}$. Combining (18) and (19) gives

$$
\begin{aligned}
\tilde{\mathcal{E}}^{(k)} &\leq \hat{\mathcal{E}}^{(K)}e^{C_{\mathcal{H}}(K-k)\Delta} + \left(\frac{e^{C_{\mathcal{H}}(K-k)\Delta} - 1}{e^{C_{\mathcal{H}}\Delta} - 1}\right)\epsilon_g \\
&\leq \left(\frac{e^{C_{\mathcal{H}}K\Delta} - 1}{e^{C_{\mathcal{H}}\Delta} - 1}\right)\epsilon_f e^{C_{\mathcal{H}}(K-k)\Delta} + \left(\frac{e^{C_{\mathcal{H}}(K-k)\Delta} - 1}{e^{C_{\mathcal{H}}\Delta} - 1}\right)\epsilon_g.
\end{aligned} \tag{22}
$$

On the other hand, we observe that

$$
\begin{aligned}
\tilde{\mathcal{E}}^{(k)} &= \left\|\tilde{\mathbf{u}}^{(k)} - \mathbf{u}(t^{(k)})\right\| \\
&\leq \left\|\tilde{\mathbf{u}}^{(k)} - \hat{\mathbf{u}}^{(k)}\right\|_2 + \left\|\hat{\mathbf{u}}^{(k)} - \mathbf{u}(t^{(k)})\right\| \\
&= \left\|\mathcal{G}_\vartheta^{(K-k)} \circ \mathcal{F}_\theta^{(K-k)}\hat{\mathbf{u}}^{(k)} - \hat{\mathbf{u}}^{(k)}\right\| + \hat{\mathcal{E}}^{(k)} \\
&\leq \left\|\mathcal{G}_\vartheta^{(K-k)} \circ \mathcal{F}_\theta^{(K-k)} - I\right\|_{L^\infty(D)} + \hat{\mathcal{E}}^{(k)} \\
&\leq \left\|\mathcal{G}_\vartheta^{(K-k)} \circ \mathcal{F}_\theta^{(K-k)} - I\right\|_{L^\infty(D)} + \left(\frac{e^{C_{\mathcal{H}}k\Delta} - 1}{e^{C_{\mathcal{H}}\Delta} - 1}\right)\epsilon_f,
\end{aligned} \tag{23}
$$

where we have used the triangular inequity for the Euclidean norm and the estimate (18). Combining (23) with (22) gives (20) and completes the proof. □

Table 3: Training time and inference time of our method and baseline method in all the experiments.

| SYSTEM | METHOD | SAMPLES | TRAINING TIME (S) | INFERENCE TIME (S) |
|---|---|---|---|---|
| DAMPED PENDULUM | OURS | 417 | 2554.8 | 0.314 |
| | BASELINE | 14400 | 749.7 | 0.296 |
| 2D NONLINEAR | OURS | 925 | 1868.4 | 0.121 |
| | BASELINE | 14400 | 444.1 | 0.129 |
| LORENZ SYSTEM | OURS | 1765 | 81473.0 | 0.403 |
| | BASELINE | 1000000 | 14936.1 | 0.384 |
| VISCOUS BURGERS' | OURS | 19683 | 68263.6 | 0.483 |
| | BASELINE | 500000 | 12137.5 | 0.488 |

# D ADDITIONAL EXPERIMENTAL RESULTS AND PERFORMANCE COMPARISONS

## D.1 EXPERIMENTS ON COMPUTATIONAL COMPLEXITY

In Table 3, we present the training time and inference time of our method and baseline method respectively in all 4 numerical experiments. It shows that our method requires more time on training. Although the proposed method has higher complexity, our method can dramatically reduce the number of needed samples, therefore, reducing the sample collection cost. This is highly important in practice, especially with real-world dynamical systems.

## D.2 ADDITIONAL EXPERIMENTAL RESULTS

We conduct experiments on the first Damped Pendulum system. Using our critical sampling and adaptive evolution operator learning method, the number of training samples needed for evolution behavior learning is reduced significantly from 14,400 to 417. Figure 8 compares the phase plots for the Damped Pendulum Equation with an arbitrarily chosen initial state $u(t = 0) = (-1.193, -3.876)$ predicted by the baseline method with 14,400 samples and our method with 417 samples. In each sub-figure, the orange line represents the ground-truth solution, and the blue dotted line shows the network prediction results. In Figure 9, the left column shows the trajectories for $(u_1, u_2) \in \mathbb{R}^2$ obtained by the baseline method with 14,400 samples and the right column shows the trajectories obtained by our method with only 417 samples. We can see that, using the proposed critical sampling method, the network can still robustly predict the long-term evolution of the system states, despite using far fewer samples for training.

Next, We experiment on the 2D Nonlinear system. In this system, we use only 925 samples to achieve similar modeling accuracy as the baseline method which has 14,400 samples. Figure 10 compares the phase plots for the 2D Nonlinear system with an arbitrarily chosen initial state $u(t = 0) = (-1.325, 1.874)$ predicted by the baseline method with 14,400 samples and our method with 925 samples. The orange line represents the ground-truth solution, and the blue dotted line shows the network prediction results. In Figure 11, the left column shows the trajectories obtained by the baseline method with 14400 samples and the right column shows the trajectories obtained by our method with 925 samples.

For the chaotic Lorenz system, our proposed method only requires 1,765 samples to achieve the same modeling error as the baseline method which requires 1,000,000 training samples. Figure 12 compares the phase plots with an arbitrarily chosen initial state $u(t = 0) = (-8, 7, 27)$ obtained by the baseline method with 1,000,000 samples and our method with 1,765 samples. In Figure 13, the left column shows the trajectories $(u_1, u_2, u_3) \in \mathbb{R}^3$ obtained by the baseline method with 1,000,000 samples and the right column shows the trajectories obtained by our method with 1,765 samples.

For the last Viscous Burgers' Equation (PDE) system, our proposed critical sampling method has drastically reduced the number of samples from 500,000 to 19,683 and achieved similar prediction accuracy. Figure 14 shows the predicted system states at different time obtained by the baseline method with 500,000 samples and our method with 19,683 samples, respectively. The orange line shows the ground-truth states at different time. Figure 15 shows the network predictions for system states on $t \in [0, 2]$ under two different initial conditions. Both methods show accurate prediction performance under those two initial conditions, but our method requires much fewer samples.

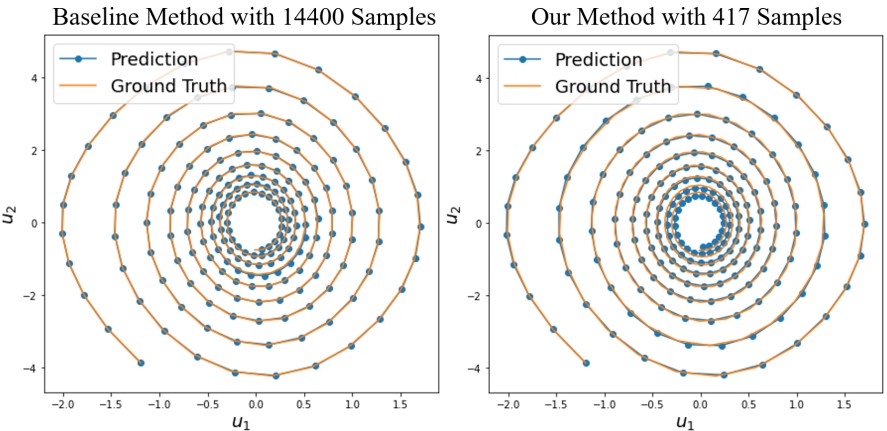

Figure 8: Phase plots for the Damped Pendulum Equation with initial state $u(t = 0) = (-1.193, -3.876)$. Left: baseline method with 14400 samples; Right: our method with 417 samples.

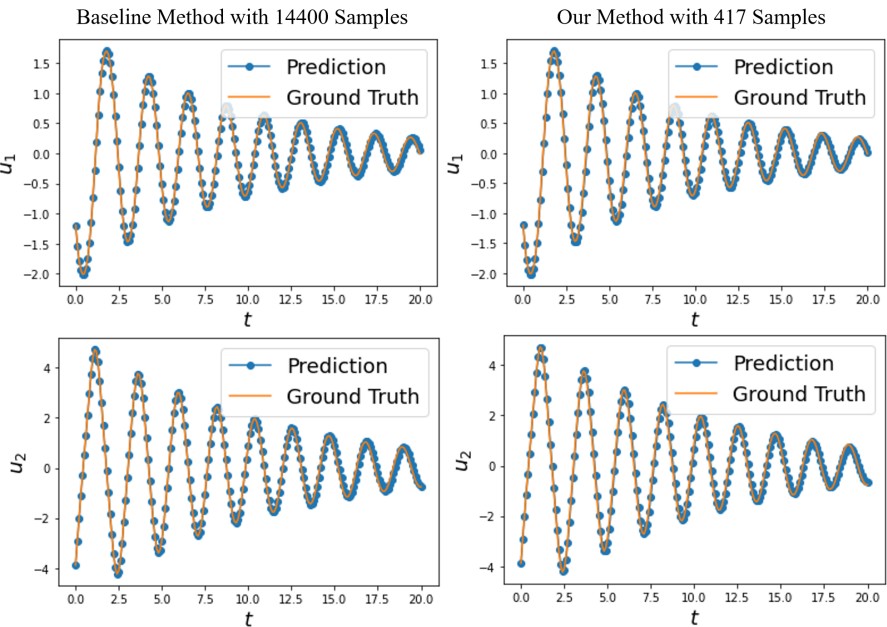

Figure 9: Trajectories for the Damped Pendulum Equation with initial state $u(t = 0) = (-1.193, -3.876)$. Left column: baseline method with 14400 samples; Right column: our method with 417 samples.

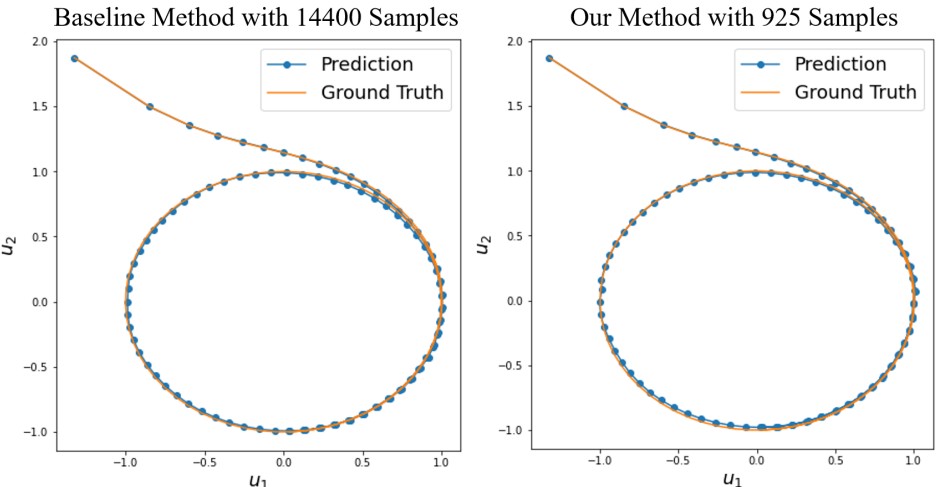

Figure 10: Phase plots for the 2D Nonlinear Equation with initial state $u(t = 0) = (-1.325, 1.874)$. Left: baseline method with 14400 samples; Right: our method with 925 samples.

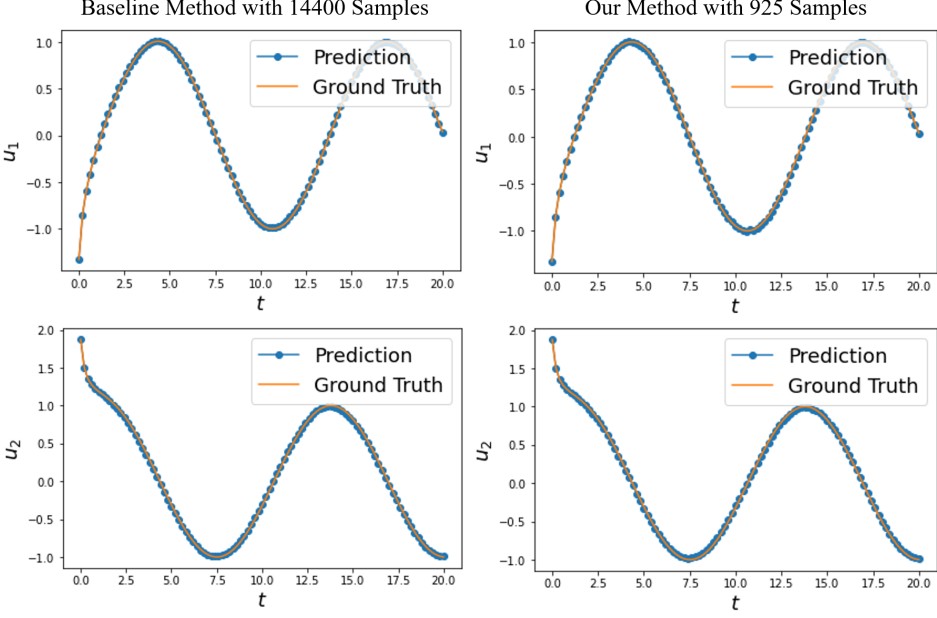

Figure 11: Trajectories for the 2D Nonlinear Equation with initial state $u(t = 0) = (-1.325, 1.874)$. Left column: baseline method with 14400 samples; Right column: our method with 925 samples.

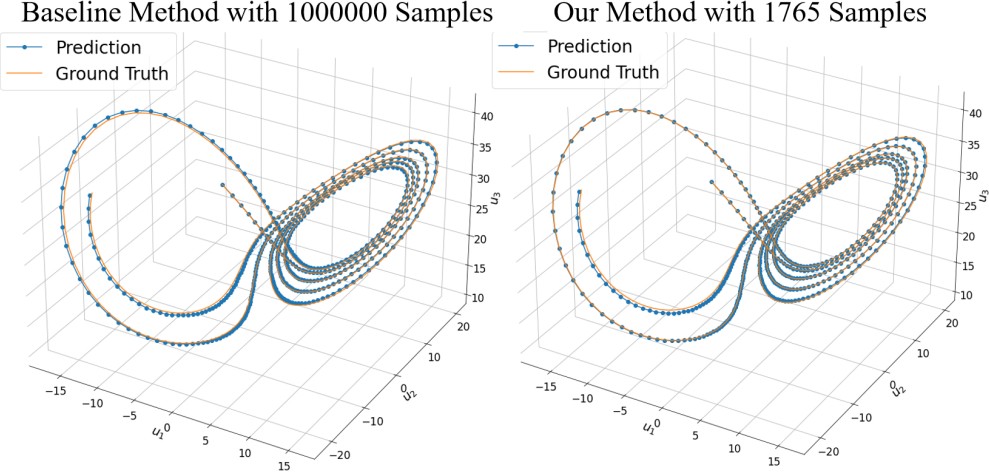

Figure 12: Phase plots for the Lorenz System with initial state $u(t = 0) = (-8, 7, 27)$. Left: baseline method with 1000000 samples; Right: our method with 1,765 samples.

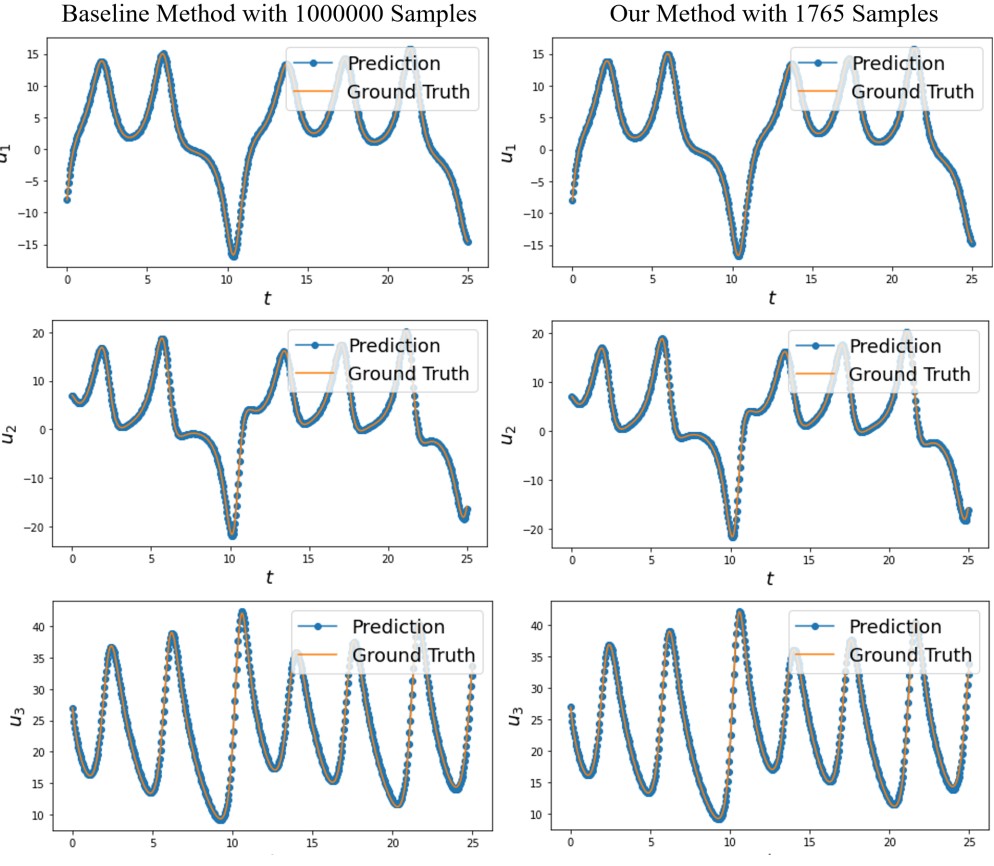

Figure 13: Trajectories for the Lorenz System with initial state $u(t = 0) = (-8, 7, 27)$. Left column: baseline method with 1000000 samples; Right column: our method with 1,765 samples.

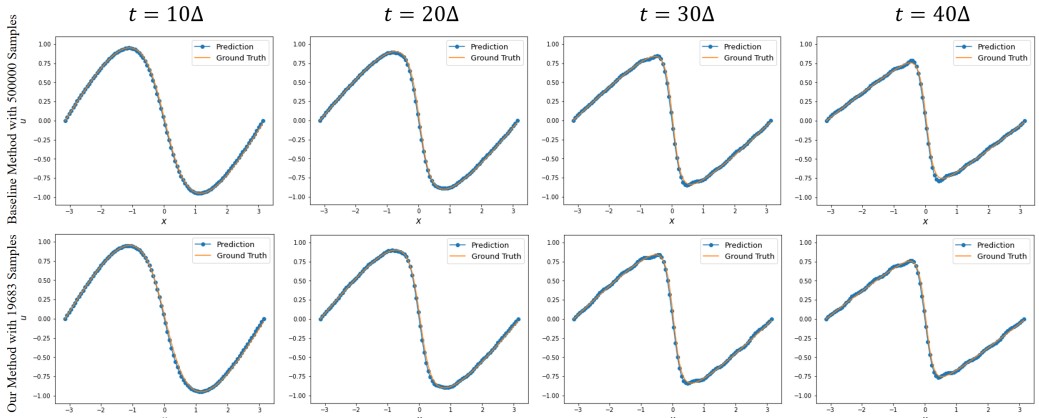

Figure 14: Viscous Burgers' Equation: comparison of the ground-truth solution and the learned network prediction at different time. Top row: baseline method with 500000 samples; Bottom row: Our method with 19683 samples.

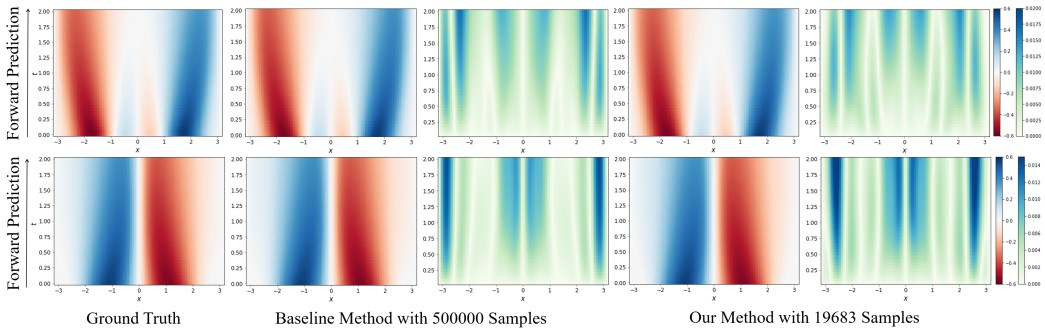

Figure 15: Top row: Comparison of ground-truth solution, baseline prediction and proposed method prediction on $t \in [0, 2]$ under initial condition $u = 0.370 \sin(x) - 0.169 \sin(2x) - 0.175 \sin(3x)$ of Viscous Burgers' Equation system. Bottom row: Comparison of ground-truth solution, baseline prediction and proposed method prediction on $t \in [0, 2]$ under initial condition $u = -0.430 \sin(x) - 0.219 \sin(2x) + 0.017 \sin(3x)$ of Viscous Burgers' Equation system.

