# OpenReview forum: "Critical Sampling for Robust Evolution Behavior Learning of Unknown Dynamical Systems"
_ICLR.cc/2023/Conference — Submitted to ICLR 2023_

### Official Review · Reviewer_6jX7 · 2022-10-25

**Confidence:** 3
**Correctness:** 2
**Technical Novelty And Significance:** 2
**Empirical Novelty And Significance:** 2
**Recommendation:** 3

**Clarity, Quality, Novelty And Reproducibility:**

The problem considered is new but in my opinion it is not significant. I did not try to reproduce the results but the parameter configurations are mentioned and one should be able to reproduce the results. The presentation can be improved, there are frequent grammatical errors but the logical flow of the paper is clear.

**Strength And Weaknesses:**

**Major Weaknesses**
* The literature review is incomplete, many other methods for learning ODEs are not mentioned. Examples are HMMs, and linear and nonlinear state space models (LDS, SLDS, fLDS, SVAE, RNNs, etc.).
* The original motivation for the presented method is that the existing models require large sample sizes to learn the dynamics and converge, however supporting figures are not accompanied. In time series and dynamical systems specifically, the critical sample size depends on many factors including the initial condition, sampling frequency, dynamical regime, etc. Further investigation is needed to address under what conditions a new method for critical sampling is helpful.
* It is unclear how can the presented method be applied in practice for physical systems. Even if we know what the critical (high error) samples are, it's often the case that we cannot simply change the state of the system towards those samples. The results are only shown in the simulations, questioning whether the method is applicable to real physical systems or not.
* Above the methods section, 4 contributions are elaborated. Contributions 1 and 2 are not independent and can be listed under the same number. Contribution 3 is not new, there is a large list of papers learning forward-backward models, i.e. simultaneously learning the forward and inverse models. In addition, the connections to HMM models are not explained, since the inference algorithms in HMM and LDS consist of a forward-backward loop, it's important to explore whether there is a connection. Importantly, HMM and LDS can handle noise, whereas in my understanding the presented model works in the noiseless regime (which is usually not the case for the real physical systems). Regarding contribution 4, I'm not quite convinced that this is fully addressed through empirical studies. The systems that are considered are very specific, comparisons are done against a specific baseline and many other models are not considered. The sampling frequency is not chosen according to the best practices.
* The systems considered in this paper are all autonomous, this limits the scope of the presented models as there are many physical systems that aren't fully autonomous. In addition, the systems considered are low-dimensional. Since the high error regions are determined by sampling a grid in the state space, in the high dimensional cases it's unclear that the method scales properly.
* The finding about the reciprocal error being correlated with the network modeling error is highly dependent on the underlying system, properties of the underlying flow field, learning algorithm, initial sample, dynamical regime, etc. It's unclear whether this finding holds for other physical systems.

**Minor Issues**
* The visualizations can be dramatically improved. The plots are Python's basic plots and the fonts are very small. The plots can be made more professionally.
* What does the following mean in the second paragraph after the methods section?
> The solution evolution is uniquely defined without bifurcation
* In the first paragraph on page 3, "effective" should be changed to "effectively".
* There is a grammar error in the following:
> Learning the local spatial dynamics. The dynamical system may exist a highly nonlinear and complex behavior in the whole spatial domain ...

## Post Rebuttal (Round 1)
I thank the authors for their detailed responses and new additions to the paper. Below I reply to specific points raised by the authors in response to my criticisms.

> Response: In the revised paper, we have included a separate section for related work (Section 2), and included a review of these statistical modeling/learning methods. It should be noted that, in this work, our focus is to study the problem of data-driven modeling of unknown dynamical systems whose behaviors are controlled by their underlying governing equations. These methods focus on statistical learning or modeling of more generic system observation data.

One interpretation of the models I mentioned such as HMMs and linear and nonlinear state space models (LDS, SLDS, fLDS, SVAE, RNNs, etc.) is that they are approximating an underlying dynamical system or ODE using a certain function family determined by the model. For example, in the case of LDS this function family is linear ($x_{t+1} = Ax_t$) whereas for SLDS this function family is piecewise linear and for RNN it can be a generic nonlinear function parameterized by a neural network $x_{t+1} = f(x_t)$. In this sense, I don't see a difference between the articulated model and these existing frameworks and I don't find the reason convincing enough to exclude a comparison with these models.


> Response: Thanks for pointing out this. In existing work on data-driven modeling of dynamic systems, they need large sets of training samples to successfully train the deep neural network to achieve a reasonably accurate approximation of the system evolution trajectories. Table 1 of the revised paper summarizes the number of needed training samples for the method to achieve the desired model prediction error. For example, for the first Damped Pendulum system, the baseline method needs 14400 samples to achieve an average prediction error of 0.0263 so that the predicted system trajectory is fairly close to the true trajectory.

As I mentioned in my original reply, intuitively the number of samples needed depends on many factors, including but not limited to the sampling frequency, dimension, and dynamical regime (e.g. chaotic, stationary, stable, etc). Since the main motivation for this work is that models require large sample sizes, this should be illustrated more thoroughly through empirical results on multiple systems when best practices are used (e.g. optimal sampling frequency and models with lower sampling requirements). A vanilla comparison with an ad hoc method where no consideration of the best practices does not sound like a convincing baseline, to begin with.

> Response: We really appreciate these insightful comments! We totally agree with you that in practical systems, it is not straightforward to change the state of the system towards those samples. Here are our thoughts and responses to address the issues that you pointed out ...

I was expecting to see some results on physical systems in response to this question. If this is applicable only to simulation I don't find the contribution significant enough for publication. If this is in fact applicable to physical systems (as the authors argue it is) results must be accompanied.

> Response: ... Sorry for the confusion. Our idea and method are quite different from the HMM and LDS. Specifically, in our work, we construct a multi-step reciprocal prediction network structure, which performs K-step forward prediction followed by K-step backward prediction by concatenating K forward prediction networks plus K backward prediction networks together. Interestingly, we find that this multi-step reciprocal prediction error obtained from this prediction loop is highly correlated with the model prediction error.

I understand the motivation and the presented method and how they are different from LDS and HMM. I apologize for the unclear question and I hope this clarifies the question further. Regardless of the motivation of your paper, the algorithm that you've presented has forward and backward parts where you run the model forward through the forward mapping and then backward through the inverse mapping and evaluate the error and backpropagate through the parameters of the model. Indeed you use the error to efficiently sample from the dynamical system for the next iterations and hence the active learning framework. Now my question is, the forward-backward algorithm appears in other time series models as well (such as Kalman filter, LDS, HMM, etc.)? It would be interesting to see if there are connections between the two sets of algorithms.


> Summary of Responses ...

I read the responses from other reviewers as well and I appreciated their input about the work. I do think however that some of the points that I raised remain concerning and unanswered and they are independent of the points raised by other reviewers. In summary, I think the empirical results do not fully support the claims in the paper and more systematic empirical evaluation is needed for backing up the arguments. Specifically, considering other models for comparisons, using best practices as the baseline, and considering underlying dynamical systems with different characteristics (dynamical regime, dimension, presence of noise, etc.) is critical for the claims to hold. Given this, I tend to keep my score as is.


## Post Rebuttal (Round 2)

I thank the authors for further explanations, in the following I wish to clarify my standpoint and the reasoning behind my score further.

**Regarding the motivation of the paper,** it is mentioned that learning ODEs using deep networks require large sample sizes and critical sampling helps with reducing the sample size. While I think that this statement is true in general when a baseline model and vanilla sampling schemes are used for comparison, I think it remains unclear under what circumstances it's worthwhile to use the proposed framework. There are ad-hoc methods for reducing the sample size. For example, one could subsample the input data using a frequency given by the auto-correlation time of the time series. Or one can run a clustering and subsample within a cluster.

I think the answer to this question depends on many factors, some of which are listed below.

It depends on the underlying model. If the underlying deep network utilizes linear functions, the sample size required for learning the dynamics is as low as the dimension of the system. While the linear model is just an example, it shows that certain deep learning function spaces (determined by the neural network architecture, choice of nonlinearity, number of nodes, and regularization) will inherently require fewer data sizes to be trained compared to others. If the effect of inductive biases of the function space on the sample size is not systematically studied it is not possible for a practitioner to decide if this framework will be useful for their case or not.

It depends on the dynamic properties of the data. For example, if the data is sampled from an attractor type or stationary ODE, intuitively, fewer samples should be needed to learn the dynamics compared to a chaotic attractor (e.g. Lorenz model). It's important to show under what regimes the claims hold so that the practitioners are able to determine if the method is applicable to their case or not.

**Regarding the significance of this work,** I wish to clarify that I do not think that every paper needs to have experimental results in it. I have given high scores to papers with no figures and no data before and I do not question the importance of having empirical studies on simulations and theoretical studies. However, I do believe that if a method is developed mainly for practical settings, accompanying it with real data will make it stronger. In my opinion, the methodological novelty of this paper is limited. Leaning forward and backward functions existed before. The main novelty comes from the empirical finding that the multi-step error is correlated with the ground truth error, hence utilizing it during sampling will help reduce the error faster than vanilla sampling. Given that the novelty lies in this empirical observation, at the very least a systematic investigation of the conditions under which the claim holds is required.

Given that in the simulated cases the constraints imposed by the computational resources are less severe and we can train deep learning models in a short time, the main merit of this work lies in experimental settings where we are time and budget limited. Given the context I provided, unless some results on real-world data are not shown (by that I certainly do not mean that the authors should do the experiments themselves) I do believe that the scope of this work is limited. In addition, noise is an inseparable component of real-world data, in the presence of which learning the forward and backward model might be more challenging and the critical sampling framework might not hold.

If we accept that the scope is limited, and we base the judgment only on the simulation results, then more controls are needed to show that if we account for time complexity and other factors, then the presented framework has better inductive biases and achieves better test/generalization error. Creating a grid of points and running the forward-backward model for a few time points is a time-consuming process. To bring the comparisons to the same footing, one should consider the time spent on the optimization and not just the number of samples used for training. Additionally, deep learning models are proven useful mainly in high-dimensional settings. There is no systematic study of how much of the claims hold in higher dimensions and the feasibility of applying the method to higher dimensional settings in case of evaluating the error on a grid and training the forward and backward functions. I think that for the acceptance of an empirical paper in ICLR, a solid empirical investigation (including the ones I mentioned above) is necessary.



## Post Rebuttal (Round 3)

> Response: Thanks for pointing out this! It should be noted that, in this paper, we are learning deep neural network models of unknown dynamical systems ...

The auto-correlation is easily estimated from the data, it is the correlation of a signal with a delayed copy of itself as a function of delay. There is no need for the equations to be known. Regarding clustering, one can think of clustering as a way to sparsify the data samples. Start from a large sample size, run a simple clustering algorithm such as k-means and use the cluster centroids or data points that are close to cluster centroids for training a neural network. These are the simplest ad-hoc algorithms to reduce the data size, but more complex ones can also be considered. My point is, in your work you have a data selection process, which is to evaluate the error on a grid of samples and choose the high-error data points for training. This data selection process should be compared to other data selection processes such as the ones I mentioned above as opposed to the vanilla fixed-interval sampling scheme used in the comparisons to motivate the paper.


> Response: Thanks for this insightful comment! We totally agree with you that the number of needed training samples does depend ...

> Response: We agree with you that experiments on real-world data will be very helpful. We will follow your comment to ...

I cannot base my judgment on the results that I have not seen. Once the new results are included I will re-assess the paper accordingly.


> Response: We totally agree with you that learning different ODEs would require different amounts of data. In our problem, ...

This needs to be carefully evaluated and investigated. I understand that in the problem statement you are assuming that the governing equations are not known. But ground truth studies can investigate under what dynamical regimes (with known equations) the method performs better than others.

> Response: Thanks for pointing out this! In this paper, we study the important problem of selecting ...

Given that in the simulated environment the proposed method takes longer to run compared to baseline methods, now I firmly believe that the main merit of the paper is in the real-world scenarios where data collection is challenging and costly. In the simulated setting, one is not concerned with the computational resources of using a larger data sample for training. In fact, this just shows another example of the no-free-lunch-theorem. You are using the computational resources in a different way (computing multi-step error for a grid of samples) compared to using the computational resources for processing batches of data.


**Summary Of The Paper:**

The paper studies the problem of critical sampling for a dynamical system by sampling from the high error regions of the state space. Since the underlying system and hence the error is not known in the region where data does not exist, the authors propose to use a surrogate error given by the reciprocal prediction error of the model. The model effective consists of two parts, a forward prediction module to predict $x_{t+1}$ from $x_t$ and a backward module to predict $x_t$ from $x_{t+1}$. The model is trained both from samples and also a reciprocal error measuring how well the prediction is going forward using the forward module for $K$ time steps and then going backward using the backward module for $K$ time steps. Some theoretical results and empirical simulation results are shown for ODEs and PDEs showing that the proposed model outperforms the baseline.

**Summary Of The Review:**

I am not convinced about the motivations behind developing this technique, whether the proposed methods can be used in real physical scenarios, and conditions under which this method can be useful and outperform other models.

---

> ### Author Response · Authors · 2022-11-18
> **Response to Reviewer 6jX7 (Part 4/4)**
>
> **Comment**: “*The finding about the reciprocal error being correlated with the network modeling error is highly dependent on the underlying system, properties of the underlying flow field, learning algorithm, initial sample, dynamical regime, etc. It's unclear whether this finding holds for other physical systems*.”
>
> **Response**: We really appreciate these valuable comments! First, we do agree with you that our finding is obtained from experiments, depending on the underlying systems, which will have different properties of flow fields, initial conditions, dynamical regimes, etc. However, we do observe that, the deep neural network is a highly effective approach for modeling and approximating the system behavior. Our multi-step reciprocal prediction error and its correlation with the network model prediction error are computed and evaluated locally at every sample location in the state space. Within a small neighborhood, the system can be well approximated by simple functions, whose behavior can be better understood. Our theoretical analysis provides a better understanding of the correlation. Therefore, we do believe this finding is important and useful. In our future work, motivated by your comments, we shall study this further for a wider range of systems with different configurations.
>
> **Comment**: “*The visualizations can be dramatically improved. The plots are Python's basic plots and the fonts are very small… In the first paragraph on page 3, ‘effective’ should be changed to ‘effectively’ ...  There is a grammar error*...”
>
> **Response**: Thanks for your careful review of our paper! In the revised paper, we have followed your comments and corrected these.
>
> **Comment**: “*What does the following mean in the second paragraph after the methods section? ‘The solution evolution is uniquely defined without bifurcation’*.”
>
> **Response**: It meant for autonomous systems the future state is uniquely determined by its current state, and there is no bifurcation case that two trajectories are evolved from the same initial state. Your comment helped us to realize that this sentence can be confusing to readers, and therefore, we have removed it in the revised paper.
>
> **Summary of Responses**:
>
> We sincerely thank you for your careful review of our paper and your insightful comments! We believe the considered question is not trivial, as agreed by the other two reviewers:
> + *This paper studied an interesting and important problem*;
> + *The problem of strategically choosing a limited training set for dynamical systems is important*;
> + *The approach in this paper seems novel and creative*.
>
> We hope that our responses have addressed your comments. If you have additional comments or concerns, please let us know and we will be more than happy to answer.
>
> Best regards,
>
> Authors

---

> ### Author Response · Authors · 2022-11-18
> **Response to Reviewer 6jX7 (Part 3/4)**
>
> **Comment**: “*Importantly, HMM and LDS can handle noise, whereas in my understanding the presented model works in the noiseless regime (which is usually not the case for the real physical systems*).”
>
> **Response**: Thanks for pointing out this! Yes, as you said, the HMM and LDS algorithms can handle noise. In your comment, you mentioned that the presented model works in the noiseless regime. We are not sure what type of noise you are mentioning. We try to answer your question based on our best understanding.
>
> In our method, when we learn or approximate the system state evolution operator using a deep neural network. As we know, deep neural network learning and approximation is a statistical regression process, which can handle a certain amount of noise in the samples. For example, in our experiment on the first Damped Pendulum system, we added 2-10% of Gaussian noise onto the measurements or training samples, we found that the prediction error of the learned model changes very little, from 0.0263 to 0.0271.
>
> **Comment**: “*The systems that are considered are very specific, comparisons are done against a specific baseline and many other models are not considered*.”
>
> **Response**: Thanks for pointing out this! In this paper, we consider data-driven modeling of unknown dynamical systems, which is a new and exciting application of deep learning methods to understand real-world systems. Unlike machine learning tasks for computer vision where large benchmark datasets, such as ImageNet or MS-COCO datasets, are available, our task involves specific dynamical systems. We follow the same evaluation procedure used in existing research, for example those reviewed in Section 2, to evaluate the performance of our proposed method on specific examples of dynamical systems. All our example systems are taken from benchmark tests for learning unknown dynamical systems.
>
> **Comment**: “*The systems considered in this paper are all autonomous, this limits the scope of the presented models… In addition, the systems considered are low-dimensional… in the high dimensional cases it's unclear that the method scales properly*.”
>
> **Response**: Thanks for this valuable comment! Yes, in this work, we assume that the system is autonomous, whose behavior can be completely characterized by its evolution operator. Certainly, we agree with you that there are also many non-autonomous systems. In the literature, for example, in the following paper, the evolution-learning type methods have been extended from the autonomous case to non-autonomous ones, especially when the system variables that cause the non-autonomous behavior are observable. In our future work, we shall extend our work to non-autonomous systems by incorporating these variables into the network learning process.
>
> [3] Tong Qin, Zhen Chen, John D. Jakeman, and Dongbin Xiu. Data-driven learning of nonautonomous systems. *SIAM Journal on Scientific Computing*, 43(3):A1607–A1624, 2021.
>
> Please note that the PDE system (Example #4) is an “infinite-dimensional” dynamical system. Its exact evolution operator is defined on an “infinite-dimensional” space, and the approximate evolution operator is in a 9-dimensional modal space.
>
> As we can see from Table 1, our proposed method becomes more and more important in high-dimensional space. Our four example dynamical systems are seemingly simple equations. However, the evolution dynamics of those systems are NOT simple. The basic setup of our work is that the equations are unknown, and we only have the data samples, from which we learn the (not simple) evolution dynamics. For example, the Lorenz system of Example #3 has extremely complicated chaos dynamics, whose evolution prediction is a well-known challenge.

---

> ### Author Response · Authors · 2022-11-18
> **Response to Reviewer 6jX7 (Part 2/4)**
>
> **Comment**: “*It is unclear how can the presented method be applied in practice for physical systems. Even if we know what the critical (high error) samples are, it's often the case that we cannot simply change the state of the system towards those samples…. Regarding contribution 4, I'm not quite convinced that this is fully addressed through empirical studies*.”
>
> **Response**: We really appreciate these insightful comments! We totally agree with you that in practical systems, it is not straightforward to change the state of the system towards those samples. Here are our thoughts and responses to address the issues that you pointed out:
>
> (1) In this work, as in existing work on data-driven modeling of unknown dynamical systems, the target system is simulated, so we do not have problems in controlling the system state. We can choose the initial state of the system and start the simulation to collect the sample.
>
> Currently, simulations play an important role in scientific research, for example, fluid dynamics, turbulence, ocean fluid dynamics, molecular dynamics, cardiovascular fluid dynamics, and weather forecasting. Often, to obtain high-precision simulations of the system, the computational cost is very high. Therefore, within the context of simulation, our proposed method is very useful by dramatically reducing the number of needed training samples for deep neural network modeling.
>
> (2) In many real-world systems, they often have system controllers which are able to steer the system towards the desired system state or within the small neighborhood of the target system state. Or, in many scenarios, for example, the ocean dynamic system, we can use existing simulation models to predict the space-time regions or conditions when and where the desired system states will occur, and then schedule the critical sample collection tasks. In this case, our proposed method can be used as an important guidance tool for sample collection. For some complex systems, the exact change of the system state may not be trivial. In this case, we shall investigate how the system state control impacts the critical sampling and system modeling performance
>
> We have added this discussion in the conclusion section (Section 5) of the revised paper.
>
> **Comment**: "*Contributions 1 and 2 are not independent and can be listed under the same number. Contribution 3 is not new, there is a large list of papers learning forward-backward models, i.e. simultaneously learning the forward and inverse models. In addition, the connections to HMM models are not explained, since the inference algorithms in HMM and LDS consist of a forward-backward loop, it's important to explore whether there is a connection*."
>
> **Response**: Thanks for your valuable comments! In the revised paper, we have revised this section to better summarize the unique contributions of our work. We follow your suggestion to combine Contributions 1 and 2.
>
> Sorry for the confusion. Our idea and method are quite different from the HMM and LDS. Specifically, in our work, we construct a multi-step reciprocal prediction network structure, which performs K-step forward prediction followed by K-step backward prediction by concatenating K forward prediction networks plus K backward prediction networks together. Interestingly, we find that this multi-step reciprocal prediction error obtained from this prediction loop is highly correlated with the model prediction error.
>
> This finding of strong correlation allows us to estimate the prediction error. This solves an important problem for deep neural network-based learning and prediction: how to model and estimate the prediction error of deep neural networks, especially for dynamic system behavior prediction. We believe this contribution is important, and uniquely different from the forward-backward process in the HMM and LDS methods.

---

> ### Author Response · Authors · 2022-11-18
> **Response to Reviewer 6jX7 (Part 1/4)**
>
> Dear Reviewer:
>
> We are very grateful for your time and efforts on careful review of our paper and your valuable comments, which helped us to improve the manuscript! In the following, we provide detailed responses to your valuable comments.
>
> **Comment**: “*The literature review is incomplete, many other methods for learning ODEs are not mentioned. Examples are HMMs, and linear and nonlinear state space models (LDS, SLDS, fLDS, SVAE, RNNs, etc.*)”
>
> **Response**: In the revised paper, we have included a separate section for related work (Section 2), and included a review of these statistical modeling/learning methods. It should be noted that, in this work, our focus is to study the problem of data-driven modeling of unknown dynamical systems whose behaviors are controlled by their underlying governing equations. These methods focus on statistical learning or modeling of more generic system observation data.
>
> **Comment**: “*The original motivation for the presented method is that the existing models require large sample sizes to learn the dynamics and converge, however supporting figures are not accompanied*.”
>
> **Response**: Thanks for pointing out this. In existing work on data-driven modeling of dynamic systems, they need large sets of training samples to successfully train the deep neural network to achieve a reasonably accurate approximation of the system evolution trajectories. Table 1 of the revised paper summarizes the number of needed training samples for the method to achieve the desired model prediction error. For example, for the first Damped Pendulum system, the baseline method needs 14400 samples to achieve an average prediction error of 0.0263 so that the predicted system trajectory is fairly close to the true trajectory.
>
> **Comment**: “*The critical sample size depends on many factors including the initial condition, sampling frequency, dynamical regime, etc. Further investigation is needed to address under what conditions a new method for critical sampling is helpful*.”
>
> **Response**: We really appreciate your insightful comment! In this paper, our research problem setup is: in data-driven modeling of unknown dynamical systems using deep neural networks, existing methods need large sets of training samples to successfully learn the network models, especially for high-dimensional systems. In practical systems, it is often costly to collect samples. How can we dramatically reduce the number of needed training samples while achieving the same model prediction error?
>
> Our paper had two unique and important contributions. First, we made an interesting finding: we introduce the multi-step reciprocal prediction error and find that it is highly correlated with the model prediction error which is hard to estimate since the ground-truth values are not available. Second, based on this finding, we developed an adaptive critical sampling method and a spatial dynamics network for sample augmentation from this small set of critical samples, which can dramatically reduce the number of needed real samples to be collected while achieving the same target model prediction error.
>
> In our experiments, we follow the same settings as many other data-driven system modeling papers to perform the simulation and experiments. For example, in the baseline method, when collecting samples for the training data, the initial conditions are randomly or uniformly sampled in the state space. We also chose the same regions to perform the system simulations as existing papers, for example, those in the following.
>
> [1] Tong Qin, Kailiang Wu, and Dongbin Xiu. Data driven governing equations approximation using deep neural networks. *Journal of Computational Physics*, 395:620–635, 2019.
>
> [2] Kailiang Wu and Dongbin Xiu. Data-driven deep learning of partial differential equations in modal space. *Journal of Computational Physics*, 408:109307, 2020.

---

> ### Author Response · Authors · 2022-12-11
> **Further Response to Reviewer 6jX7 (Part 3/3)**
>
> **We sincerely wish that you could kindly consider our responses to your comments and re-evaluate our paper from the following perspective**: as with any other scientific paper, this paper has its specific problem formulation. All papers and methods have their own limitations. However, if the paper DOES develop a new idea or has a new finding, and it provides clear experimental evidence to support this new idea or finding, the paper can be accepted. We believe the new finding in this paper, as agreed by the other reviewers, is a significant contribution and well supported by experimental results. We do appreciate your valuable and insightful comments, which point out the limitations and additional research issues that need to be addressed in our future work.
>
> Best regards,
>
> Authors

---

> ### Author Response · Authors · 2022-12-11
> **Further Response to Reviewer 6jX7 (Part 2/3)**
>
> **Comment**: "*I was expecting to see some results on physical systems in response to this question. If this is applicable only to simulation, I don't find the contribution significant enough for publication. If this is in fact applicable to physical systems (as the authors argue it is) results must be accompanied*."
>
> **Response**: Thanks for your comment! We agree with your comments, but, it should be noted that it is very challenging for us or beyond our capability to construct experimental platforms to sense and control physical systems, such as ocean surveillance, fluid dynamic systems, and other dynamical systems in this paper. Since these systems can be well modeled by differential equations, it is reasonable and scientifically valuable to evaluate our methods using data generated by these models. Also, note that the main focus of this paper is to demonstrate that our method is able to significantly reduce the number of needed training samples for deep neural networks. Certainly, if we are able to implement the proposed method in the field dynamical system, the impact will be more significant.
>
> **Comment**: "*I understand the motivation and the presented method and how they are different from LDS and HMM. … Indeed you use the error to efficiently sample from the dynamical system for the next iterations and hence the active learning framework. Now my question is, the forward-backward algorithm appears in other time series models as well (such as Kalman filter, LDS, HMM, etc.)? It would be interesting to see if there are connections between the two sets of algorithms...*"
>
> **Response**: Thanks for this valuable comment! Sorry for the misunderstanding. As you said, we did run the model forward through the forward mapping and then backward through the inverse mapping and evaluated the error. This error is called the multi-step prediction error. We did NOT back-propagate through the parameters of the model. Instead, we found that this multi-step reciprocal prediction error is highly correlated with the network modeling error, which cannot be obtained without knowing the ground-truth. This finding allows us to schedule the critical samples guided by the multi-step reciprocal prediction error. So, besides the forward and backward prediction, our method or contribution is much different from existing statistical modeling methods, such as Kalman filters, LDS, or HMM, since they solve different problems: their aim for statistical modeling of the system, while our method aims for quantifying the modeling errors for adaptive sampling. In the final paper, we will follow your comment to include this discussion about their connections.
>
> **Comment**: “*I read the responses from other reviewers as well and I appreciated their input about the work. I do think however that some of the points that I raised remain concerning and unanswered and they are independent of the points raised by other reviewers. In summary, I think the empirical results do not fully support the claims in the paper and more systematic empirical evaluation is needed for backing up the arguments. Specifically, considering other models for comparisons, using best practices as the baseline, and considering underlying dynamical systems with different characteristics (dynamical regime, dimension, presence of noise, etc.) is critical for the claims to hold. Given this, I tend to keep my score as is*.”
>
> **Response**: While the other two reviewers have agreed that this paper has made unique contributions from the machine learning and deep neural network modeling perspective since it addressed a very important problem: when we learn a deep neural network to predict the behavior of an unknown system, how do we know the prediction error or network modeling error since the ground truth is not available. In this paper, we construct the multi-step reciprocal error and find that it is highly correlated with the network modeling error. This finding along with the critical sampling method is a valuable and novel contribution to machine learning and deep neural networks.
> Although the experiments are structured differently from other statistical modeling papers, our experiments follow the procedures of existing papers and demonstrate that our method is able to significantly reduce the number of needed training samples to achieve the same network modeling error. So, we believe these experiments are convincing. Certainly, in our future work, to demonstrate its impact in practical systems, we shall follow your suggestion to deploy our method in field systems and evaluate its performance.

---

> ### Author Response · Authors · 2022-12-11
> **Further Response to Reviewer 6jX7 (Part 1/3)**
>
> Dear Reviewer:
>
> We are very grateful for your time and efforts on careful review of our paper and your valuable comments.
>
> **Comment**: "*One interpretation of the models I mentioned such as HMMs and linear and nonlinear state space models (LDS, SLDS, fLDS, SVAE, RNNs, etc.) is that they are approximating an underlying dynamical system or ODE using a certain function family determined by the model. … In this sense, I don't see a difference between the articulated model and these existing frameworks and I don't find the reason convincing enough to exclude a comparison with these models*."
>
> **Response**: Thanks for pointing out this! The main contribution of this paper is that our critical sampling method is able to dramatically reduce **the number of needed training samples for deep neural networks** to achieve the same network modeling error. So, the focus of our experiments is to demonstrate the effectiveness of this critical sample method instead of the efficiency of the underlying network models. Our method is specifically designed for deep neural network modeling of unknown dynamical systems. We realize that our method does not work with HMMs and linear and nonlinear state space models (LDS, SLDS, fLDS, SVAE, RNNs, etc.) This is the reason that we could not include them for performance comparison. But, in our related work, we have included these methods for review in the related work section and discuss the connection between them.
>
> **Comment**: "*Since the main motivation for this work is that models require large sample sizes, this should be illustrated more thoroughly through empirical results on multiple systems when best practices are used (e.g. optimal sampling frequency and models with lower sampling requirements). A vanilla comparison with an ad hoc method where no consideration of the best practices does not sound like a convincing baseline, to begin with*."
>
> **Response**: Thanks for your comment! We have followed your comment to conduct additional experiments to demonstrate the performance of our method for different sampling frequencies and target error rates. As we can see from the following table, our method is able to significantly reduce the number of needed training samples for the deep neural networks when modeling the unknown dynamical systems.
>
> | Baseline Samples | Prediction Error | Our Samples | Prediction Error | Ratio |
> | :--------------: | :-----------------------: | :---------: | :------------------: | :---: |
> | *Damped Pendulum* |                           |             |                      |       |
> |       3600       |          0.00695          |     496     |       0.00673        | 7.26  |
> |       6400       |          0.00254          |     625     |       0.00250        | 10.24 |
> | 10000 |          0.00057          |     825     |       0.00045        | 12.12 |
> | 14400 |          0.00037          |     925     |       0.00035        | 15.57 |
> |   *2D Nonlinear*   |                           |             |                      |       |
> |       3600       |          0.12803          |     225     |       0.12706        | 16.00 |
> |       6400       |          0.07459          |     297     |       0.07370        | 21.55 |
> | 10000 |          0.04370          |     333     |       0.04345        | 30.03 |
> | 14400 |          0.02630          |     417     |       0.02411        | 37.53 |
>
> Also, we would like to point out our experimental design is valid and also solid: for the same sample deep neural network, using our method, we can dramatically reduce the number of samples needed to successfully train the deep neural network to approximate the unknown dynamical system, achieving the same approximation error. We have also provided convincible evidence that the multi-step reciprocal prediction error is highly correlated with the network model error. So, these experiments can well support the idea and contributions made in this paper. We followed the same procedure as in existing papers to evaluate our data-driven modeling method.

---

> ### Author Response · Authors · 2022-12-12
> **Discussion Continued - Response to Reviewer 6jX7 (Part 2/2)**
>
> **Comment**: “*Regarding the significance of this work... However, I do believe that if a method is developed mainly for practical settings, accompanying it with real data will make it stronger. In my opinion, the methodological novelty of this paper is limited…. Given the context I provided, unless some results on real-world data are not shown (by that I certainly do not mean that the authors should do the experiments themselves) I do believe that the scope of this work is limited. In addition, noise is an inseparable component of real-world data, in the presence of which learning the forward and backward model might be more challenging and the critical sampling framework might not hold*.”
>
> **Response**: We agree with you that experiments on real-world data will be very helpful. We will follow your comment to include additional performance evaluations. For example, we will use the data from existing seminal papers, such as those listed below, to further evaluate the performance of our method. We will include these new comparison results in the final paper.
>
> [1] Brunton, S. L., Proctor, J. L., & Kutz, J. N. (2016). Discovering governing equations from data by sparse identification of nonlinear dynamical systems. *Proceedings of the National Academy of Sciences*, *113*(15), 3932-3937.
>
> [2] Rudy, S. H., Brunton, S. L., Proctor, J. L., & Kutz, J. N. (2017). Data-driven discovery of partial differential equations. *Science Advances*, *3*(4), e1602614.
>
> [3] Lusch, B., Kutz, J. N., & Brunton, S. L. (2018). Deep learning for universal linear embeddings of nonlinear dynamics. *Nature Communications*, *9*(1), 1-10.
>
> **Comment**: “*Leaning forward and backward functions existed before. The main novelty comes from the empirical finding that the multi-step error is correlated with the ground truth error, hence utilizing it during sampling will help reduce the error faster than vanilla sampling*.”
>
> **Response**: Thanks for your comments! We agree with that the forward and backward functions existed before. However, our method is not about forward and backward functions. The main finding of our paper is that the multi-step reciprocal prediction error obtained from the cascades of forward and backward evaluation operator networks is highly correlated with the network modeling error. This finding is interesting and novel. It allows us to estimate the network prediction error, which cannot be obtained without knowing the ground-truth.
>
> **Comment**: “*To bring the comparisons to the same footing, one should consider the time spent on the optimization and not just the number of samples used for training. Additionally, deep learning models are proven useful mainly in high-dimensional settings. There is no systematic study of how much of the claims hold in higher dimensions and the feasibility of applying the method to higher dimensional settings in case of evaluating the error on a grid and training the forward and backward functions*.”
>
> **Response**: Thanks for pointing out this! In this paper, we study the important problem of selecting critical samples for effective deep neural network learning unknown dynamical systems. We believe this problem is very important. As for the optimization time, in Section D.1 of our paper, we have included a discussion on the complexity of our method.
>
> We agree that our four example dynamical systems are seemingly simple equations (low-dimensional). However, the evolution dynamics of those systems are NOT simple. The basic setup of our work is that the equations are unknown, and we only have the data samples, from which we learn the (not simple) evolution dynamics. For example, the Lorenz system of Example #3 has extremely complicated chaos dynamics, whose evolution prediction is a well-known challenge. Please note that the PDE system (Example #4) is an "infinite-dimensional" dynamical system. Its exact evolution operator is defined on an "infinite-dimensional" space, and the approximate evolution operator is in a 9-dimensional modal space. In Section 4.1 of the revised paper, we have explained this.
>
> Again, we really appreciate your time and efforts to provide valuable feedback on our paper!
>
> Best regards,
>
> Authors

---

> ### Author Response · Authors · 2022-12-12
> **Discussion Continued - Response to Reviewer 6jX7 (Part 1/2)**
>
> Dear Reviewer 6jX7,
>
> We sincerely thank you for your detailed responses and appreciate this discussion with you, which helped us improve the paper.
>
> **Comment**: "*Regarding the motivation of the paper …While I think that this statement is true in general when a baseline model and vanilla sampling schemes are used for comparison, I think it remains unclear under what circumstances it's worthwhile to use the proposed framework. There are ad-hoc methods for reducing the sample size. For example, one could subsample the input data using a frequency given by the auto-correlation time of the time series. Or one can run a clustering and subsample within a cluster*."
>
> **Response**: Thanks for pointing out this! It should be noted that, in this paper, we are learning deep neural network models of **unknown** dynamical systems. In other words, specific models or statistics about the system, including the auto-correlation statistics, are NOT available in our study. Therefore, current mathematical or statistical methods for dynamic sampling do not apply or work. The clustering subsample method you proposed would first require access to a sufficiently large set of measurement samples, group them into clusters, and then dynamically select sub-samples from each cluster. This procedure and logic do not fit our problem, which assumes that the large set of measurement samples is difficult and expensive to acquire. Our critical sampling and learning method does not require any prior samples. It is able to discover a very small set of critical samples in a progressive manner and learn the deep neural network system model effectively.
>
> **Comment**: "*…It depends on the underlying model. If the underlying deep network utilizes linear functions, the sample size required for learning the dynamics is as low as the dimension of the system. While the linear model is just an example, it shows that certain deep learning function spaces (determined by the neural network architecture, choice of nonlinearity, number of nodes, and regularization) will inherently require fewer data sizes to be trained compared to others. If the effect of inductive biases of the function space on the sample size is not systematically studied it is not possible for a practitioner to decide if this framework will be useful for their case or not*."
>
> **Response**: Thanks for this insightful comment! We totally agree with you that the number of needed training samples does depend on the specific structure and configuration of the deep neural network. We have followed your important suggestion to evaluate the performance of our critical sampling methods on networks with different structures and configurations. Our experiments have shown large reductions (about 5-15 times) in needed training samples. We will include these new results in the final paper.
>
> **Comment**: “*It depends on the dynamic properties of the data. For example, if the data is sampled from an attractor type or stationary ODE, intuitively, fewer samples should be needed to learn the dynamics compared to a chaotic attractor (e.g. Lorenz model). It's important to show under what regimes the claims hold so that the practitioners are able to determine if the method is applicable to their case or not*.”
>
> **Response**: We totally agree with you that learning different ODEs would require different amounts of data. In our problem, the governing equations of the system are totally unknown. We only have access to a few measurement data, from which it is very difficult, or even impossible, to identify the ODE types (simple or complicated, stationary or chaotic).
>
> Actually, no matter the underlying unknown governing equations are complicated or not, our proposed method is always helpful for the effective discovery of critical samples. When the system is simple, the multi-step reciprocal prediction error will be small, then fewer samples will be selected. When the system is complicated, the error will be larger, and more critical samples will be discovered.

---

### Official Review · Reviewer_mGbk · 2022-10-25

**Confidence:** 3
**Clarity, Quality, Novelty And Reproducibility:** The paper is clearly written and well…
**Correctness:** 4
**Technical Novelty And Significance:** 3
**Empirical Novelty And Significance:** 4
**Recommendation:** 8

**Strength And Weaknesses:**

This is an interesting paper, which studied an important problem in dynamical system modeling. The methodology is well presented and easy to follow. I also want to commend on the numerical results and their figures, which are easy to be interpreted and demonstrate promising performance. The finding of strong correlation between the network modeling error and the multi-step reciprocal prediction error is super interesting and it is also very smart to use the multi-step reciprocal prediction error to approximate the desired network modeling error.

**Summary Of The Paper:**

This paper studied an interesting and important problem and tried to answer the question: given an unknown dynamical system, how to predict its evolution behavior accurately with minimum number of samples and how to select these samples. To be specific, the paper introduces a multi-step reciprocal prediction network, where a forward evolution network and a backward evolution network are designed to learn and predict the temporal evolution behavior in the forward and backward time directions, respectively. In addition to that, the paper also proposed a joint spatial-temporal evolution network which extends the framework to the space. The numerical result is appealing and demonstrates promising performance compared to baseline approach.


**Summary Of The Review:**

See above

---

> ### Author Response · Authors · 2022-11-18
> **Response to Reviewer mGbk**
>
> Dear Reviewer:
>
> Many thanks for your valuable time to review our paper!
>
> **Comment**: “*This paper studied an interesting and important problem….The numerical result is appealing and demonstrates promising performance compared to baseline approach*.”
>
> **Response**: We really appreciate your positive review of our paper and encouraging comments.
>
> **Comment**: “*The finding of strong correlation between the network modeling error and the multi-step reciprocal prediction error is super interesting and it is also very smart to use the multi-step reciprocal prediction error to approximate the desired network modeling error*.”
>
> **Response**: Thank you very much for summarizing the unique contributions of our work! As we know, one central challenge in deep neural network learning is that we do not have an effective way to measure or estimate the prediction error. It is a very interesting finding that, within the context of data-driven modeling or learning of the dynamical systems, the multi-step reciprocal prediction error is highly correlated with the actual model prediction error, which allows us to develop an adaptive critical sampling scheme. We really appreciate your positive review of our paper and the encouraging comments!
>
> Thank you again for your positive recommendation of our paper!
>
> Best regards,
>
> Authors

---

### Official Review · Reviewer_C4XT · 2022-10-29

**Confidence:** 3
**Correctness:** 3
**Technical Novelty And Significance:** 4
**Empirical Novelty And Significance:** 4
**Recommendation:** 8

**Clarity, Quality, Novelty And Reproducibility:**

Novelty: as mentioned above, this seems like a novel and creative approach to this problem. However, the literature review could be improved and I don't agree that "this work is one of the first efforts to address this challenge." I found [A] and [B] quickly as examples, but saw many other papers that are addressing this problem, sometimes with a particular application area of dynamical systems in mind.

The literature review and vocabulary could be improved.
1. I found the paragraph from the Introduction beginning with "In recent years, data-driven discovery of the governing equations of physical systems from measurement data has emerged as an important area of research. There are two major approaches that have been explored" confusing. Firstly, I had a hard time understanding the distinction of the two approaches. Is the core distinction between learning the equations vs. learning the operator? If that's the point you're trying to make, then I think it's confusing to specify "classic sparse regression methods and modern neural networks" in the introductory sentence of the first approach and "train a deep neural network" in the introductory sentence of the second approach, since that's more specific than a lot of the literature, including the citations that follow. Also, "data-driven discovery of the governing equations" sounds to me like at the end you have equations that you can write down. In that case, I would include references for symbolic regression. I also wouldn't include citations for approaches that end in a black-box model without equations because that's a different task (perhaps "approximation" of the governing equations or surrogate modeling rather than "discovery" of the equations). Maybe you want to use broader language than "data-driven discovery of the governing equations" for this paragraph or split off the black-box approaches into a different paragraph? (I later found that there is a section in the appendix that repeats some of this, and the wording there is more clear.)
2. Since this is a machine learning conference, it would be helpful to use machine learning terminology by mentioning that this is an "active learning" problem. This is not mentioned until the very end when the paper says it's "related" to active learning methods. I think that this "Further Discussion" paragraph at the end that mentions the relationship to a few other methods would make more sense in the introduction when other related work is discussed. I also don't see any citations for other papers that work on the same problem of choosing good samples for learning a dynamical system. Note that some related work would be under names like "optimal experiment design" and "system identification." A couple of recent examples from a quick search are [A] and [B].


[A] Mania, Horia, Michael I. Jordan, and Benjamin Recht. "Active Learning for Nonlinear System Identification with Guarantees." J. Mach. Learn. Res. 23 (2022): 32-1.

[B] Huang, Yu, et al. "Physics-Coupled Spatio-Temporal Active Learning for Dynamical Systems." IEEE Access (2022).



Clarity:

I found the "Sample augmentation based on local spatial dynamics" section confusing. If points are successfully predicted by the spatial dynamics network, they are added to the sample set. What's the definition of being successfully predicted by the spatial dynamics network? Does that mean that you already know the future state for these points? If so, then doesn't this invalidate the goal of being able to know which samples to add without already knowing the ground-truth?

Algorithm 1 (in the Appendix) adds some clarity, but I found it still vague. For example, is the multi-step reciprocal prediction error averaged across \bar{S_F^n}? How big is the "large set of samples" V_F? The main paper references adding points if they are successfully predicted by the spatial dynamics network. Is that incorporated somewhere in Algorithm 1? Within "generate a large set of samples"?
Until the appendix, I couldn't find any reference to how the final errors were defined as in Table 1, Figure 5, etc.. Based on Section A.3 in the appendix, I assume that those errors in Table 1 & Figure 5 are the errors on the 1-5 trajectories mentioned in this section, and evolving forward much longer (the intervals in this section)? Is that correct? Are those trajectories ones that are not in the training set? I think the errors shown in the main paper should have some definition, since this is confusing, even if the full details are in the appendix.

I found it confusing when I got to the definition of S^n_F, which uses J_n. I eventually figured out that n is used in two ways: dimensionality of the system, and an index for the step in the active learning process. It would be helpful to choose a different letter and to explain J_n. (This is not defined until the appendix.)

The fonts in the figures are often too small to read.

"With this, we are able to perform dynamic selection of critical samples from regions with high network modeling errors and develop an adaptive sampling-learning method for dynamical systems based on the spatial-temporal evolution network." Since you have two ways of adding more training samples, how did you balance the two? (The interaction of these two ideas was somewhat explained once I got to the Appendix, Section A.4, but I'm still confused. I think this should be more clear in general, but also that there should be some explanation of this in the main paper.)

In Section A.3: a sentence abruptly ends: "is used and."


Quality:

"This number is empirically chosen since it is needed for the network to achieve a reasonably accurate and robust learning performance." How did you define the number of samples needed by the baseline method? There would be a tradeoff between number of samples and error, just like with your method. Did you test how few samples you could use while reaching a target error? This is important to clarify in order to see if the claims are accurate.

The paper emphasizes that the problem of needing more training data is much worse in higher dimensions. However, the examples in this paper are primarily 2-3 dimensional. At first, I thought that the Burgers equation example would be high-dimensional (the dimensionality of the spatial discretization). However, in the appendix, we learn that the PDE is tranformed into a lower-dimensional problem (I think 9-dimensional.) I think this should be stated in the main paper for fairness.


Reproducibility: The code and data are not shared. As described above, there are many aspects of this method that I found unclear, so I would not be able to reproduce this paper.

Of note: I did not check the proofs.

**Post Rebuttal**

Thanks for your responses! I think that most of my concerns were addressed, and I will raise my score. I look forward to trying this method on my problems after it is published.


**Strength And Weaknesses:**

Strengths:

I think the problem of strategically choosing a limited training set for dynamical systems is important, and I'm glad to see more work done in this direction. The approach in this paper seems novel and creative to me.

If I understand correctly, the reported errors are on very long trajectories, whereas the temporal evolution model is trained on just one step. Long-term prediction of dynamical systems is difficult, so it's impressive to see comparisons with long-term errors.

Weaknesses:

I think that the writing could be improved in terms of clarity, reproducibility, and the writing on related work. (I have more details on that below.) Also, if I understand correctly, the errors reported are on 1-5 trajectories, which is very low in my experience. I would like to see errors averaged across a larger number of trajectories to see if the network generalizes well. I have additional comments in the next box.

**Summary Of The Paper:**

This paper presents an approach to predicting a dynamical system with a spatio-temporal neural network and an active learning method for choosing a limited number of training samples to collect. They are able to greatly reduce the number of samples needed to predict a dynamical system.

**Summary Of The Review:**

I think that this paper is interesting (important problem and novel approach). However, the writing needs quite a bit of work, and I have some clarification questions that would help me understand how strong the claims are. As mentioned above, I would also like to see errors averaged over more trajectories that were not used for training.

---

> ### Author Response · Authors · 2022-11-18
> **Response to Reviewer C4XT (Part 2/2)**
>
> **Comment**: “*Is the multi-step reciprocal prediction error averaged across $\bar{S_F^n}$? How big is the ‘large set of samples’ $V_F$… How did you define the number of samples needed by the baseline method*?”
>
> **Response**: In our critical sampling scheme, the multi-step reciprocal prediction error is computed for every sample of the augmented training set  $\bar{S_F^n}$. It should be noted this augmented training set includes the original small set of collected samples and the large set of augmented samples or interpolated samples by the spatial dynamics network. As found in the paper, this multi-step reciprocal prediction error is highly correlated with the actual model prediction error. Once we have the model prediction error at every sample, we can then determine which regions or subsets of samples (interpolated ones) need to collect more real samples, instead of being interpolated from real samples. We have added a detailed explanation in Section 3.2 of the revised paper.
>
> The augmented training set includes a very small set of real samples and the large set of augmented or interpolated samples. In our experiment, we set its size to be the same as the training set of the baseline method. For the baseline method, the size of the training set is chosen so that the desired average model prediction error is reached. For example, the desired prediction error for the first Damped Pendulum system is 0.0263. Certainly, as you said, there would be a tradeoff between the number of samples and error. For the baseline method, if we add more training samples, the average model prediction error is reduced. We then compare our method against the baseline method: to achieve the same desired prediction error, our critical sampling method can significantly reduce the number of needed real samples.
>
> **Comment**: “*I found it confusing when I got to the definition of $S^n_F$, which uses $J_n$. I eventually figured out that $n$ is used in two ways*.”
>
> **Response**: We really appreciate your careful review! In the revised paper, we have added the definition of $S^n_F$ and $J_n$ (now $m$) and their further explanations in Section 3.1. In the revised paper, we now use different symbols, $n$ and $m$ for the indices of dimensions and iterations, respectively. Please see the revised paper.
>
> **Comment**: “*The examples in this paper are primarily 2-3 dimensional. At first, I thought that the Burgers equation example would be high-dimensional …. this should be stated in the main paper for fairness*.”
>
> **Response**: Thanks for pointing out this! We agree that our four example dynamical systems are seemingly simple equations (low-dimensional). However, the evolution dynamics of those systems are NOT simple. The basic setup of our work is that the equations are unknown, and we only have the data samples, from which we learn the (not simple) evolution dynamics. For example, the Lorenz system of Example #3 has extremely complicated chaos dynamics, whose evolution prediction is a well-known challenge. Please note that the PDE system (Example #4) is an “infinite-dimensional” dynamical system. Its exact evolution operator is defined on an “infinite-dimensional” space, and the approximate evolution operator is in a 9-dimensional modal space. In Section 4.1 of the revised paper, we have explained this.
>
> **Comments**: “*In Section A.3: a sentence abruptly ends: ‘is used and’ …The fonts in the figures are often too small to read*.”
>
> **Response**: Thank you very much for the careful review! In the revised paper, we have corrected these following your comments.
>
> We hope that our responses have addressed your concerns. If you have additional comments or concerns, please let us know and we will be more than happy to answer. Again, we really appreciate your time and efforts to provide valuable feedback on our paper.
>
> Best regards,
>
> Authors

---

> ### Author Response · Authors · 2022-11-18
> **Response to Reviewer C4XT (Part 1/2)**
>
> Dear Reviewer:
>
> We really appreciate your thorough review of our paper and your valuable comments! We also thank you for the positive and encouraging comments like “*the problem of strategically choosing a limited training set for dynamical systems is important*” and “*the approach in this paper seems novel and creative*”. In the following, we provide detailed responses to your valuable comments.
>
> **Comment**: "*The errors reported are on 1-5 trajectories, which is very low in my experience… the errors shown in the main paper should have some definition… Is that correct? Are those trajectories ones that are not in the training set*?”
>
> **Response**: Thanks for this valuable comment! In the revised paper, we have updated experimental results, and reported the errors on 50 trajectories for all systems with different initial conditions. These initial conditions and trajectories are not in the training set. We can see that, to achieve the same average prediction errors, the ratios of sample size reduction are almost the same. Please see Table 1 and Figure 5 for more details. Following your comment, we have also added the definition of error to the revised paper in Table 1.
>
> **Comment**: "*I found [A] and [B] quickly as examples, but saw many other papers that are addressing this problem, sometimes with a particular application area of dynamical systems …Since this is a machine learning conference, it would be helpful to use machine learning terminology by mentioning that this is an active learning problem*.”
>
> **Response**: Thanks for pointing out this! In the revised paper, we have provided a more comprehensive review of related work on this topic, especially those on active learning, optimal experiment design, and system identification. Following your comment, we have re-introduced our paper from the active learning perspective to better fit the paper into the conference.
>
> **Comment**: “*I had a hard time understanding the distinction of the two approaches, …. perhaps 'approximation' of the governing equations or surrogate modeling rather than 'discovery' of the equations*.”
>
> **Response**: Thank you for your insightful comments and helpful suggestions. We agree with your comments. Following your suggestions, we have included a separate section for the related work in the revised paper and carefully revised the paragraph you mentioned. In particular, we use broader language “data-driven modeling” to replace the previous “data-driven discovery”. We have also included two seminal references on symbolic regression and clarified that the first approach for data-driven modeling aims to learn the mathematical expressions or formulas of the unknown equations, while the second approach seeks to approximate the evolution operator of the underlying equations. We wish that our revised paper has addressed your concerns. In case we missed any important references in the literature review, please let us know so that we can make it more complete.
>
> **Comment**: “*I found the ‘Sample augmentation based on local spatial dynamics’ section confusing*…”
>
> **Response**: Sorry for the confusion. In this revised paper, we have explained this more clearly. Specifically, we use the spatial dynamics network to learn the spatial dynamics of the system. It acts like an interpolator in the spatial domain. From samples collected at a relatively small set of locations, using this network, we can interpolate a large set of samples at other spatial locations. With this augmented large set of samples, we can successfully train the temporal evolution network. We have added this explanation in the revised paper. Please see Section 3.3 of the revised paper for details.

---

> ### Author Response · Authors · 2022-12-08
> **Message from Authors**
>
> Dear Reviewer C4XT,
>
> Thank you very much for your kind review and insightful comments. Since the deadline for the discussion period is approaching, we would appreciate it if you could let us know whether there are any further questions about the paper or the response. We are looking forward to further discussions. If all your concerns have been resolved, it would be much appreciated if you may raise the rating of our work.
>
> We greatly appreciate your valuable time and great efforts in improving our paper!
>
> Best regards,
>
> Authors

---

> ### Author Response · Authors · 2022-12-12
> **Thank you very much!**
>
> Dear Reviewer C4XT,
>
> We sincerely thank you for your time and efforts in reviewing our paper and appreciate your detailed and valuable comments for improving our paper! Thanks for improving the paper rating and your positive recommendation of our paper!
>
> Best regards,
>
> Authors

---

### Decision · Program_Chairs · 2023-01-20

**Decision:**

Reject

**Justification For Why Not Higher Score:**

See weaknesses above.

**Justification For Why Not Lower Score:**

N/A

**Metareview: Summary, Strengths And Weaknesses:**

Please note that in this meta-review, the AC is also acting as an extra reviewer in this particular paper per discussion with PCs as the reviews were irreconcilably divergent in their evaluation.

Summary: This paper is on the theme of adaptively selecting the most informative critical states to learn a unknown autonomous dynamical system from. The literature on this topic is vast: it relates to active learning, experiment design and exploration techniques in model-based reinforcement learning. The main technical contribution is to define a "multi-step reciprocal prediction error" that is found to be highly correlated with true prediction error, and can be used for sample selection. The multi-step reciprocal prediction is based on simultaneously learning a forward and a backward evolution operator, and noting that the composition must behave like the identity; even for k-fold compositions.

Strengths: The paper is clearly written, easy to understand and tackles an important problem since sample efficiency is bottleneck for estimating physical systems from data. The empirical results on 4 systems show that relative to a couple of baselines, the proposed critical state selection method gives similar or lower prediction errors with a fraction of training samples.

Weaknesses:
(1) Methods developed in the model-based RL literature are natural alternatives, but there is no comparison reported. These include uncertainty aware approaches such as PILCO (Deisenroth and Rassmussen) and probabilistic ensembles with trajectory sampling (PETS) and references therein, which can be easily applied to autonomous problems using uncertainty estimates as a guide for sample selection.

(2) It is not clear what properties the forward and backward networks must have in order to yield a mismatch error that correlates with the true network error. For example, if the backward evolution operator is a perfect inverse (e.g., simply the reverse ODE associated with a Neural ODE based forward model), then the mismatch error at any initial condition will be zero.  This suggests that the robustness of the method may be sensitive to the parameterizations of the forward and backward models, but there is no empirical insights offered.

(3) For a practical method and a problem of interest in several domains, the empirical demonstration is extremely weak as only small toy 2D/3D systems are considered. In Fig 5, the baseline method should also be benchmarked with varying number of samples to see if the performance mostly saturates quickly. It is unclear how effective the approach will be for increasing data dimensionality. There are other limitations: no real physical systems are considered and only autonomous case is studied and hence the scope comes across as far more restricted than, for example, related methods proposed in RL/Robotics literature and demonstrated to be working on real applications. More thorough analysis on larger or more realistic systems with more comprehensive coverage of baselines would strengthen this paper, but in its current form, it is significantly below the acceptance threshold.